# Homeostasis, injury, and recovery dynamics at multiple scales in a self-organizing mouse intestinal crypt

Louis Gall[1]*, Carrie Duckworth[2], Ferran Jardi[3], Lieve Lammens[3], Aimee Parker[4], Ambra Bianco[5], Holly Kimko[1], David Mark Pritchard[2], Carmen Pin[1]*

[1]Clinical Pharmacology and Quantitative Pharmacology, Clinical Pharmacology and Safety Sciences, R&D, AstraZeneca, Cambridge, United Kingdom; [2]Institute of Systems, Molecular and Integrative Biology, University of Liverpool, Liverpool, United Kingdom; [3]Preclinical Sciences and Translational Safety, Janssen, Beerse, Belgium; [4]Gut Microbes and Health Programme, Quadram Institute, Norwich, United Kingdom; [5]Clinical Pharmacology and Safety Sciences, AstraZeneca, Cambridge, United Kingdom

*For correspondence:
louis.gall@astrazeneca.com (LG);
carmen.pin@astrazeneca.com
(CP)

Competing interest: See page
13

Reviewing Editor: Mariana
Gómez-Schiavon, Universidad
Nacional Autónoma de México,
Mexico

## Abstract

The maintenance of the functional integrity of the intestinal epithelium requires a tight coordination between cell production, migration, and shedding along the crypt–villus axis. Dysregulation of these processes may result in loss of the intestinal barrier and disease. With the aim of generating a more complete and integrated understanding of how the epithelium maintains homeostasis and recovers after injury, we have built a multi-scale agent-based model (ABM) of the mouse intestinal epithelium. We demonstrate that stable, self-organizing behaviour in the crypt emerges from the dynamic interaction of multiple signalling pathways, such as Wnt, Notch, BMP, ZNRF3/RNF43, and YAP-Hippo pathways, which regulate proliferation and differentiation, respond to environmental mechanical cues, form feedback mechanisms, and modulate the dynamics of the cell cycle protein network. The model recapitulates the crypt phenotype reported after persistent stem cell ablation and after the inhibition of the CDK1 cycle protein. Moreover, we simulated 5-fluorouracil (5-FU)-induced toxicity at multiple scales starting from DNA and RNA damage, which disrupts the cell cycle, cell signalling, proliferation, differentiation, and migration and leads to loss of barrier integrity. During recovery, our in silico crypt regenerates its structure in a self-organizing, dynamic fashion driven by dedifferentiation and enhanced by negative feedback loops. Thus, the model enables the simulation of xenobiotic-, in particular chemotherapy-, induced mechanisms of intestinal toxicity and epithelial recovery. Overall, we present a systems model able to simulate the disruption of molecular events and its impact across multiple levels of epithelial organization and demonstrate its application to epithelial research and drug development.

## Editor's evaluation

The proposed model makes an important contribution to the field, allowing a better understanding of the formation and response dynamics of the intestinal crypt through the effective evaluation of healthy, disease and treatment conditions. The authors provided convincing evidence of the validity of their model and their conclusions.

## Introduction

The intestinal tract is lined by a cellular monolayer which is folded to form invaginations, called crypts, and protrusions, called villi, in the small intestine. The stem cell niche is formed by inter-mingling Paneth and stem cells located at the base of the crypt (*Barker et al., 2007*). Stem cells

divide symmetrically, forming a pool of equipotent cells that replace each other following neutral drift dynamics (*Lopez-Garcia et al., 2010*). Continuously dividing stem cells at the base of the crypt give rise to secretory and proliferative absorptive progenitors that migrate towards the villus, driven by proliferation-derived forces (*Parker et al., 2017*). The transit-amplifying region above the stem cell niche fuels the rapid renewal of the epithelium. The equilibrium of this dynamic system is maintained by cell shedding from the villus tip into the gut lumen (*Wright and Alison, 1984*).

Epithelial cell dynamics is orchestrated by tightly regulated signalling pathways. Two counteracting gradients run along the crypt–villus axis: the Wnt gradient, secreted by mesenchymal and Paneth cells at the bottom of the crypt, and the bone morphogenetic protein (BMP) gradient generated in the villus mesenchyme, with BMP inhibitors secreted by myofibroblasts and smooth muscle cells located around the stem cell niche (*Gehart and Clevers, 2019*). These two signalling pathways are also the target of stabilizing negative feedback loops comprising the turnover of Wnt receptors (*Hao et al., 2012*; *Koo et al., 2012*; *Clevers, 2013b*; *Clevers and Bevins, 2013c*) and the modulation of BMP secretion (*Büller et al., 2012*; *van den Brink et al., 2004*). Paneth cells and mesenchymal cells surrounding the niche also secrete other proliferation-enhancing molecules such as epidermal growth factor (EGF) and transforming growth factor-α (*Gehart and Clevers, 2019*). In addition, Notch signalling-mediated lateral inhibition mechanisms are essential for stem cell maintenance and differentiation into absorptive and secretory progenitors (*Gehart and Clevers, 2019*). There is also an increasing awareness of the importance of the mechanical regulation of cell proliferation through the Hippo signalling pathway interplaying with several of the key signals, such as EGF, WNT, and Notch, although the exact mechanisms are not currently fully understood (*Gehart and Clevers, 2019*).

The imbalance of this tightly orchestrated system contributes to pathological conditions, including microbial infections, intestinal inflammatory disorders, extra-intestinal autoimmune diseases, and metabolic disorders (*Chelakkot et al., 2018*). In addition, critically ill patients and patients receiving chemotherapy/radiotherapy often show severely compromised intestinal barrier integrity (*Chelakkot et al., 2018*). For instance, oncotherapeutics-induced gastrointestinal toxicity is frequently a life-threatening condition that leads to dose reduction, delay, and cessation of treatment and presents a constant challenge for the development of efficient and tolerable cancer treatments (*Stein et al., 2010*; *Saltz et al., 2000*; *Saltz et al., 2001*; *McQuade et al., 2016*). This intestinal toxicity often results from the interaction of the drug with its intended molecular target such as cell cycle proteins (*Zhang et al., 2021*) or the disruption of the cycle through DNA damage (*Helleday et al., 2008*). Multiscale models integrating our knowledge on how the epithelium maintains homeostasis and responds to injury can contribute to understand epithelial biology and quantify the risk of intestinal toxicity during drug development.

Several agent-based models (ABMs) have been proposed to describe the complexity and dynamic nature of the intestinal crypt. Early models were used as in silico platforms to study the dynamics and cellular organization of the crypt. For instance, one of the pioneering ABMs was used to study the distribution and organization of labelling and mitotic indices (*Meineke et al., 2001*). This model comprises a fixed ring of Paneth cells beneath a row of stem cells, which divide asymmetrically to produce a stem cell and a transit-amplifying cell that terminally differentiates after a fixed number of divisions. Some subsequent models are lattice-free, recapitulate neutral drift of equipotent stem cells, and describe proliferation and cell fate regulated by a fixed Wnt signalling spatial gradient, which is defined by the distance from the crypt base, with proliferating cells progressing through discrete phases of the cell cycle and showing variable duration of the G1 phase (*Pitt-Francis et al., 2009*). Further model refinements can be seen in the model of *Buske et al., 2011*, with stochastic cell growth and division time (*Buske et al., 2011*), Wnt levels defined by the fixed local curvature of the crypt and lateral inhibition driven by Notch signalling. Here, we present a lattice-free ABM that describes the spatiotemporal dynamics of single cells in the small intestinal crypt driven by the interaction of surface-tethered Wnt signals, cell–cell Notch signalling, BMP-diffusive signals, RNF43/ZNRF3-mediated feedback mechanisms, and the cycle protein network responding to the crypt mechanical environment. We show that our computational model enables the simulation of the ablation and recovery of the stem cell niche as well as of how drug-induced molecular perturbations trigger a cascade of disruptive events spanning from the cell cycle to single-cell arrest and/or apoptosis, altered cell migration and turnover, and ultimately loss of epithelial integrity.

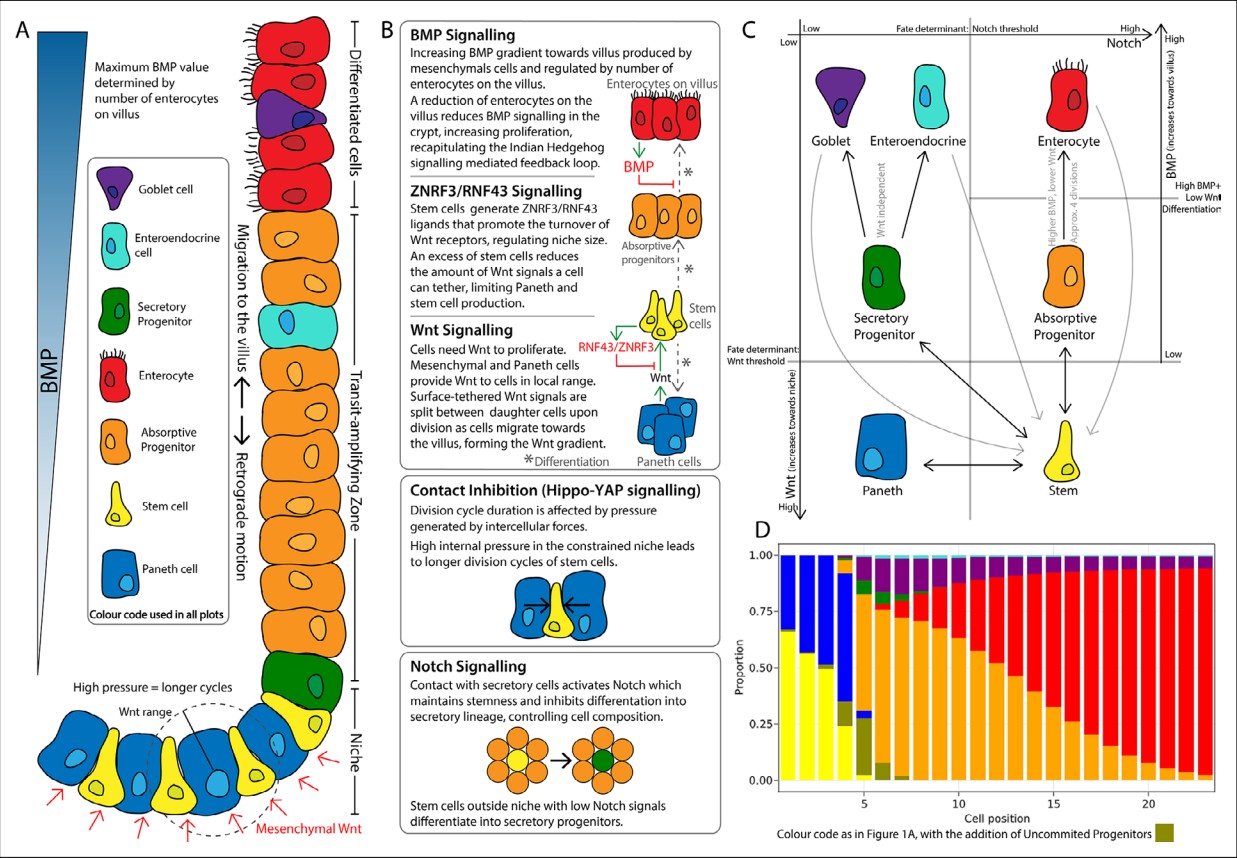

**Figure 1.** Schematics of the small intestinal crypt composition and cell fate signalling pathways included in the agent-based model (ABM). (**A**) Depiction of the crypt highlighting key signalling features and cell types in each crypt region. (**B**) Details of signalling pathways including the formation of the Wnt signalling gradient with high levels of Wnt in the stem cell niche generated by Paneth and mesenchymal cells. Intercellular pressure regulates the duration of the division cycle (YAP-Hippo pathway-mediated contact inhibition of proliferation) which impacts on the accumulation of cell surface-tethered Wnt signals. Notch signalling maintains the balance between Paneth and stem cells through lateral inhibition. A ZNRF3/RNF43-mediated feedback mechanism modulates Wnt signalling in the niche restricting the number of stem and Paneth cells. BMP signals generated by mature villus cells form a feedback loop that regulates maturation and proliferation of absorptive progenitors. (**C**) Cell fate determination. High Wnt signalling and activation of Notch are required to maintain stemness. Low Notch signalling determines differentiation into secretory fates, including Paneth cells in high Wnt signalling regions, or goblet/enteroendocrine progenitors in low Wnt regions. Absorptive progenitors develop from stem cells in low Wnt conditions and divide 3–5 times, before becoming terminally differentiated when Wnt signal levels are decreased and cells find sufficient BMP signals. (**D**) Average composition of a simulated healthy/homeostatic crypt (over 100 simulated days), showing the relative proportion of cells at each position.

## Results

### Modelling a self-organizing crypt using an ABM

We have modelled the mouse intestinal crypt as a self-organizing system where cell dynamics and cell composition arise from local interactions between single cells and the mesenchyme through signalling pathways with behaviours (proliferation, differentiation, fate decision, migration, etc.) determined largely by endogenous intracellular and intercellular interactions.

The model describes the spatiotemporal dynamics of stem cells and progenitors undergoing division cycles and responding to intercellular signalling to differentiate into Paneth, goblet, and enteroendocrine cells and enterocytes (*Figure 1A*). All cells interact physically and biochemically in the geometry of the crypt. Stem cells intermingle with Paneth cells at the bottom of the crypt and randomly replace each other. Progenitors and mature cells migrate towards the villus driven by proliferation forces (*Figure 1A*). To achieve a stable crypt cell composition under constant cell renewal dynamics, we have implemented several signalling mechanisms which include the Wnt, Notch, and BMP pathways essential for morphogenesis and homeostasis of the intestinal crypt (*Gehart and Clevers, 2019*; *Fevr et al., 2007*; *VanDussen et al., 2012*; *Pellegrinet et al., 2011*; *He et al., 2004*), the YAP-Hippo signalling pathway responding to mechanical forces and modulating contact inhibition

of proliferation (*Gjorevski et al., 2016*), and a ZNRF3/RNF43-like-mediated feedback mechanism between Paneth and stem cells to regulate the size of the stem cell niche according to experimental reports (*Hao et al., 2012*; *Koo et al., 2012*; *Farin et al., 2016*; *Figure 1B*).

The Wnt pathway is the primary pathway associated with stem cell maintenance and cell proliferation in the crypt (*Fevr et al., 2007*; *van der Flier and Clevers, 2009*). Our model implements two sources of Wnt signals described in the crypt: Paneth cells (*Sato et al., 2011*) and mesenchymal cells surrounding the stem cell niche at the crypt base (*Stzepourginski et al., 2017*). Wnt signalling is modelled as a short-range field around Wnt-emitting Paneth and mesenchymal cells with Wnt signals tethered to receptive cells as previously reported (*Farin et al., 2016*; *Clevers and Nusse, 2012*). Surface-tethered signals are split between daughter cells upon cell division (*Gehart and Clevers, 2019*; *Farin et al., 2016*), which results in a gradual depletion of tethered Wnt signals as cells divide and migrate towards the villus away from Wnt sources (*Figure 1A and B*). Notch signalling is also implemented in the model with Notch ligands expressed by secretory cells binding to Notch receptors on neighbouring cells and preventing them from differentiating into secretory fates, in a process known as lateral inhibition, that leads to a checkerboard/on-off pattern of Paneth and stem cells in the niche (*VanDussen et al., 2012*). Specifically, in our model, high Wnt and Notch signalling environments are required to maintain stemness, as reported in the literature (*Tian et al., 2015*) while under low Notch and high Wnt signalling, stem cells differentiate into secretory cells, including Paneth cells. On the other hand, Notch signalling also mediates the process of Paneth cell de-differentiation into stem cells to regenerate the niche as previously reported (*Mei et al., 2020*; *Yu et al., 2018*). Stem cells with decreased levels of Wnt signalling, usually located outside the niche, differentiate into absorptive proliferating progenitors or alternatively into secretory progenitors in the absence of Notch signals (*Figure 1C*).

In our model, mechanical stimuli, captured through the YAP-Hippo signalling pathway (*Gjorevski et al., 2016*; *Halder et al., 2012*; *Aragona et al., 2013*; *Low et al., 2014*), indirectly interact with the Notch and Wnt signalling pathways. We recapitulate YAP-mediated contact inhibition of proliferation by using cell compression to modulate the duration of the division cycle which increases when cells are densely squeezed, such as in the stem cell niche, and decreases if cell density falls, for instance, in the transit-amplifying compartment or in cases of crypt damage (*Figure 1A and B*). In agreement with experimental reports (*Pin et al., 2015*), in our model, Paneth cells are assumed to be stiffer and larger than other epithelial cells, requiring higher forces to be displaced and generating high intercellular pressure in the niche. Due to the increased mechanical pressure, cells in the niche have longer division cycles and can accumulate more Wnt and Notch signals. These premises imply that Paneth cells enhance their own production by generating Wnt signals and inducing prolonged division times, which increases stem and Paneth cell production and could lead to unlimited expansion of the niche recapitulating the phenotype seen in ZNRF3/RNF43 knockout mice (*Koo et al., 2012*; see Appendix 1, Section 1.11). To generate a niche of stable size, we implemented a negative Wnt-mediated feedback loop that resembles the reported stem cell production of RNF43/ZNRF3 ligands to increase the turnover of Wnt receptors in nearby cells (*Hao et al., 2012*; *Koo et al., 2012*; *Clevers, 2013b*; *Clevers and Bevins, 2013c*). Similarly, in our model, a number of stem cells in excess of the homeostatic value reduces cell tethering of Wnt ligands and hence inhibits Paneth and stem cell generation (*Figure 1A and B*).

The Wnt gradient in the crypt is opposed by a gradient of bone morphogenic protein (BMP) that inhibits cell proliferation and promotes differentiation (*Qi et al., 2017*). We assume that enterocytes secrete diffusing signals, resembling Indian Hedgehog signals (*Büller et al., 2012*), that induce mesenchymal cells to generate a BMP signalling gradient effective to prevent proliferative cells from reaching the villus (*Figure 1A and B*). Based on experimental evidence, we also assume that BMP activity is counteracted by BMP antagonist-secreting mesenchymal cells surrounding the stem cell niche (*McCarthy et al., 2020*). Proliferative absorptive progenitors migrating towards the villus lose Wnt during every division and eventually meet values of BMP that overcome the proliferation-inducing effect of Wnt signalling (*He et al., 2004*). We found that a homeostatic crypt cell composition is achieved when BMP and Wnt differentiation thresholds result in progenitors dividing approximately four times before differentiating into enterocytes (*Figure 1C*). In our model, the BMP signalling gradient responds dynamically to the number of enterocytes, giving rise to a negative feedback loop between enterocytes on the villus and their proliferative progenitors in the crypt that recapitulates the enhanced crypt

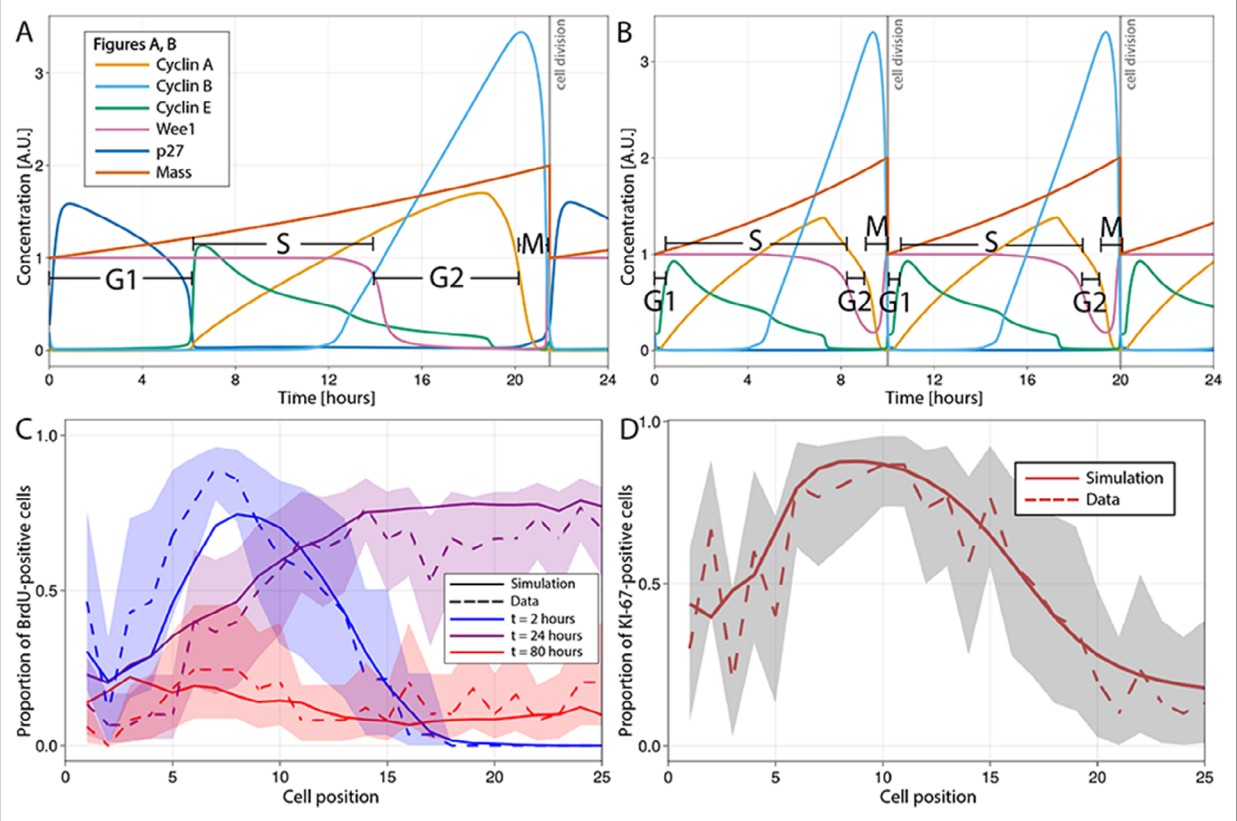

**Figure 2.** Multiscale modelling of cell division in single cells of the agent-based model (ABM). (**A**, **B**) Modelled dynamics of the main cell cycle proteins across the phases of division in each single cell over a 24 hr period, according to the cell cycle regulatory protein network model of Csikasz-Nagy (*Csikász-Nagy et al., 2006*). The protein interaction diagram can be found in the original report of Csikasz-Nagy (*Csikász-Nagy et al., 2006*). Stem cells in the crowded niche (**A**) exhibit longer cycles, up to 21.5 hr on average, with elevated levels of p27 regulating the duration of G1 and the starting of S-phase. Cells in the transit-amplifying compartment (**B**) have shorter cycles, up to 10 hr on average, due to low levels/lack of p27 expression which leads to G1 shortening and early start of the S-phase. A.U., arbitrary units. (**C**) Observed (dashed line) and simulated (solid line) proportions of BrdU-positive cells at each crypt position at 2 hr (blue), 24 hr (purple), and 80 hr (red) after a single pulse of BrdU. (**D**) Observed and simulated Ki-67-positive cells at each crypt position assuming that Ki-67 is detected in cycling cells at all phases except G1 and in any recently differentiated and arrested cells. Shadows depict the 95% confidence interval of our simulated staining results assuming that the proportion of staining cells has a beta distribution and estimating its error from experimental data.

proliferation observed after epithelial damage (*Büller et al., 2012*; *Pont and Yan, 2018*; *Sprangers et al., 2021*). For instance, a decreased number of enterocytes results in reduced production of BMP, which enables progenitor cells to divide and migrate further up the crypt before meeting BMP levels higher than the differentiation threshold.

Altogether our model describes single cells that generate and respond to signals and mechanical pressures in the crypt–villus geometry to give rise to a self-organizing crypt which has stable spatial cell composition over time (*Figure 1D*) and reproduces reported experimental data (*Buske et al., 2011*). An extended description of these modelling features is provided in Appendix 1.

## The cell cycle protein network governs proliferation in each single cell of the ABM and responds to mechanical cues

We have used the model of *Csikász-Nagy et al., 2006*, which is based on the seminal work of *Novak and Tyson, 1993*; *Novak et al., 2001*; *Novák and Tyson, 2004* and available in BioModels (*Le Novère and Csikasz-Nagy, 2006*), to recreate the dynamics of the main proteins governing the mammalian cell cycle in each single proliferative cell of the ABM. In this model, a dividing cell begins in G1, with low levels of cyclins A, B, and E and a high level of Wee1, and progresses to S-phase when cyclin E increases. S-phase ends and G2 begins when Wee1 falls. The decrease in cyclin A expression defines

the start of M-phase, while falling cyclin B implies the end of M-phase, when the cell divides into two daughter cells with half the final mass value and re-enters the cell cycle (*Figure 2A–D*).

To implement YAP-Hippo-mediated contact inhibition of proliferation, we have modified the dynamics of the proteins of the Csikasz-Nagy model to respond to mechanical cues encountered by cells migrating along the crypt. Crowded, constrained environments result in longer cycles, such as in stem cells in the niche, while decreased intercellular forces lead to shortened cycles as cells migrate towards the villus in agreement with experimental reports (*Wright and Alison, 1984*; *Marshman et al., 2002*; *Potten et al., 1997*). The shorter cycle duration in absorptive progenitors has been mainly associated with shortening/omission of G1, while the duration of S-phase is less variable (*Wright and Alison, 1984*). Using the model of *Csikász-Nagy et al., 2006*, we modulated the duration of G1 through the production rate of the p27 protein. The p27 protein has been reported to regulate the duration of G1 by preventing the activation of cyclin E-Cdk2 which induces DNA replication and the beginning of S-phase (*Morgan and Morgan, 2007*). We, hence, hypothesized that rapid cycling absorptive progenitors located in regions of low mechanical pressure outside the stem cell niche have low levels of p27, which bring forward the start of S-phase to shorten G1 (*Figure 2D*). In support of this hypothesis, it has been demonstrated that p27 inhibition has no effect on the proliferation of absorptive progenitors (*Zheng et al., 2008*; see Appendix 1 for a full description). These new features of the cell cycle model are updated dynamically and continuously to respond to changes in mechanical pressure experienced by each cell as it migrates along the crypt.

To demonstrate the performance of the model to reproduce the spatiotemporal cell dynamics and composition of a homeostatic crypt, we simulated previous published mouse experiments (*Parker et al., 2017*; *Parker et al., 2019*) comprising 5-bromo-29-deoxyuridine (BrdU) tracking (*Figure 2E*) and Ki-67 staining (*Figure 2F*). BrdU is a thymidine analogue often used to track proliferative cells and their descendants along the crypt–villus axis (*Nowakowski et al., 1989*; *Gratzner, 1982*). BrdU is incorporated into the newly synthesized DNA of dividing cells during S-phase and transmitted to daughter cells, regardless of whether they proliferate. If the exogenous administration of this molecule is discontinued, the cell label content is diluted by each cell division and is no longer detected after 4–5 generations (*Wilson et al., 2008*). To simulate the BrdU chase experiment after a single BrdU pulse, we assumed that any cell in S-phase incorporated BrdU permanently into its DNA for the first 120 min after injection of BrdU and BrdU cell content was diluted upon cell division such that after five cell divisions, BrdU was not detectable. See Appendix 1 for a complete description. The BrdU chase simulation showed that the observed initial distribution of cells in S-phase as well as division, differentiation, and migration of BrdU-positive cells over time was replicated by our model (*Figure 2E*).

Ki-67 is a protein produced by actively proliferating cells during the S-, G2-, and M-phases of the division cycle (*Sobecki et al., 2017*). Due to the time required for this protein to be catabolized (*Miller et al., 2018*), Ki-67 is also detected in quiescent or non-proliferative cells after exiting the cycle (*Miller et al., 2018*) and during G1 in continuously cycling cells (*Sobecki et al., 2017*). Our simulations assumed that Ki-67 is detected in continuously cycling cells, cells re-entering the cycle after arrest except during G1, as well as in differentiated cells that were cycling within the past 6 hr and recently drug-arrested cells. See Appendix 1 for a complete description. Similarly, we observed that the ABM-simulated spatial distribution along the crypt of Ki-67-positive cells recapitulated observations in mouse ileum (*Figure 2F*).

In summary, proliferative cells in the ABM respond to mechanical cues by adjusting the cell cycle protein network to dynamically change the duration of the cycle while migrating along the crypt. With this feature, the model replicates spatiotemporal patterns of cell proliferation, differentiation, and migration observed in mouse experiments.

## Cell plasticity/de-differentiation enables crypt regeneration following damage of the stem cell niche

Marker-based lineage-tracing studies have demonstrated numerous potential sources available for intestinal stem cell regeneration (*Hageman et al., 2020*). In line with these studies, our model assumes that cell fate decisions are reversible and both secretory and absorptive cells are able to revert into stem cells when regaining sufficient Wnt and Notch signals.

To investigate the potential of the ABM to describe and explore cell plasticity dynamics, we simulated the repeated ablation of intestinal stem cells resembling a previously published study (*Tan et al.,*

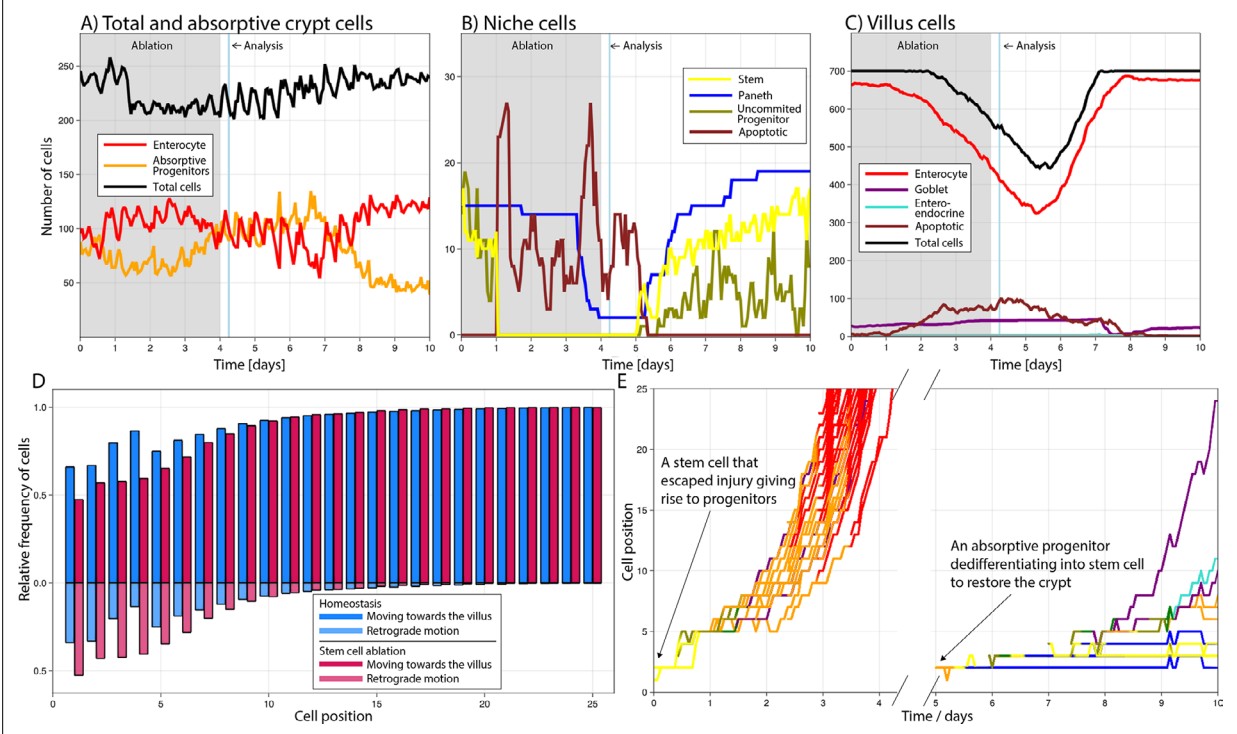

**Figure 3.** Simulated cell dynamics in the epithelium subjected to continuous ablation of stem cells for four consecutive days (grey block) resembling a previously published experiment (**A–C**) (*Tan et al., 2021*). Analysis time denotes 6 hr after ablation interruption for comparison with reported results (*Tan et al., 2021*). All cell lineages are recorded during treatment and few days after recovery of the simulated crypt for comparison with homeostasis. A simulated 3D image of a crypt in homeostasis can be found in *Figure 4A*. (**A**) shows the total number of cells, absorptive progenitors, and enterocytes in the crypt; (**B**) shows the number of Paneth, stem cells, and uncommitted progenitors, mostly found in the niche; and (**C**) shows villus cells. (**D**) Relative frequency of crypt cells moving towards the villus (darker colour) and towards the crypt base, that is, retrograde motion (lighter colour), in homeostasis (blue) and during stem cell ablation (red) at each cell position, showing increased retrograde cellular motion in the niche following stem cell ablation. (**E**) Leftmost: trajectories (cell position on crypt–villus longitudinal axis vs time) of the progeny of one stem cell, with both daughters leaving the niche and giving rise to a cascade of absorptive and secretory cells that eventually leave the crypt. Rightmost: trajectories of the progeny of an absorptive progenitor dedifferentiating into a stem cell during recovery after stem cell ablation.

*2021*). Following the experimental setup in that study, we simulated the diphtheria toxin receptor-mediated conditional targeted ablation of stem cells for four consecutive days considering that ablation was completed after the first 24 hr (*Saito et al., 2001*) and persistently inducing stem cell death during the remaining days of treatment (*Figure 3A–C*). Our simulations showed that 6 hr after the last induction, stem cells were not detected, Paneth cells decreased by 75–100% (*Figure 3B*), and the villus length was reduced by about 10–20% (*Figure 3C*) which was similar to the reported experimental findings (*Tan et al., 2021*). Simulated proliferative absorptive progenitors were indirectly affected by stem cell ablation and their decrease was followed by a reduction in mature enterocytes. The progenitors recovered after treatment interruption to later reach values above baseline when responding to the negative feedback signalling from mature enterocytes (*Figure 3A*). In our simulations, enhanced crypt proliferation was not accompanied by

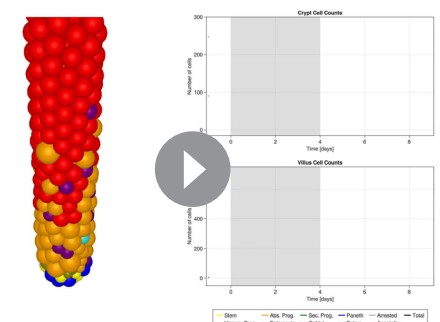

**Video 1.** Simulated cell dynamics in the epithelium subjected to continuous ablation of stem cells for four consecutive days resembling a previously published experiment (*Tan et al., 2021*). Plots depict changes in the number of cells in the crypt and villus during the simulation. Colour code of cell types is included below plots.

https://elifesciences.org/articles/85478/figures#video1

simultaneous villus recovery, which started later. *Tan et al., 2021* reported similar results with increased crypt proliferation replenishing first the crypt and not contributing immediately to villus recovery. See *Video 1* to visualize the response of the crypt.

We next studied the type of cells that were dedifferentiating during the simulated repeated ablation of stem cells and found that in agreement with experimental reports, Paneth cells (*Yu et al., 2018*), absorptive progenitors (*Tetteh et al., 2016*), and quiescent stem cells located just above the stem cell niche at the fourth cell position from the crypt base (*Tian et al., 2011*) dedifferentiated into stem cells. Specifically, from all dedifferentiated cells, about 60% were Paneth cells, 30% absorptive progenitors, and 10% secretory progenitors, which are considered quiescent stem cells as previously suggested (*Buczacki et al., 2013*). Furthermore, we used our model to explore the retrograde motion, reported using intravital microscopy (*Azkanaz et al., 2022*), of cells returning to the niche to de-differentiate into stem cells. For cells outside the niche, movement is retrograde when its velocity is negative in the *z* direction, that is, they move towards the niche across the longitudinal crypt–villus axis. For cells in the hemispherical niche, we consider a cell to move forwards, towards the villus, or backwards, towards the crypt base, if the rate of change of its polar angle is positive or negative, respectively. This implies that cells can be recorded to move backwards despite being located at the crypt base. We observed that the frequency of retrograde, or backward, movements is relatively high at low positions in a crypt in homeostasis (*Figure 3D*) and increases further after stem cell ablation, reflecting increased retrograde cellular motion as cells repopulate the niche. While in homeostasis the progeny of a stem cell generally differentiates into a cascade of absorptive and secretory progenitors that migrate towards the villus and eventually leave the crypt (*Figure 3E*). Following the interruption of stem cell ablation, during recovery absorptive progenitors return to the niche and dedifferentiate to regenerate multiple stem and Paneth cells as well as progenitors (*Figure 3E*).

Taken together, our model recapitulates cellular reprogramming of both multipotent precursors and committed progeny in the crypt and replicates the reported crypt injury dynamics following persistent ablation of stem cells (*Tan et al., 2021*).

## Disturbance of cell cycle proteins spans across scales to impact on crypt and villus organization

The model of *Csikász-Nagy et al., 2006* enables the simulation of the disruption of the main proteins governing the cell cycle in each single proliferative cell of the ABM. CDKs play important roles in the control of cell division (*Malumbres, 2014*), and the development of CDK inhibitors for cancer treatment is an active field of research (*Zhang et al., 2021*).

To explore the effect of the disruption of the cell cycle on epithelial integrity, we simulated the inhibition of CDK1 for 6 hr, every 12 hr for four consecutive days, resembling epithelial toxicity of a theoretical drug. CDK1 is reported to be the only CDK essential for the cell cycle in mammals (*Santamaría et al., 2007*). CDK1 triggers the initiation of cytokinesis by inducing the nuclear localization of mitotic cyclins A and B (*Pesin and Orr-Weaver, 2008*), and its inhibition has been proposed as a cancer therapy with potentially higher efficacy than the inactivation of other CDKs (*Diril et al., 2012*). To mimic CDK1 inhibition, we added a term to the CycA/CDK1,2 and CycB/CDK1 differential equations of the Csikasz-Nagy model (*Csikász-Nagy et al., 2006*) that strongly reduces the production of both CycA/CDK1,2 and CycB/CDK1 during the CDK1 inhibition period (*Figure 4A–E*; Appendix 1).

It has been experimentally demonstrated that the selective inhibition of CDK1 activity in cells programmed to endoreduplicate (i.e. cells that can duplicate their genome in the absence of intervening mitosis) leads to the formation of stable nonproliferating giant cells, whereas the same treatment triggers apoptosis in cells that are not developmentally programmed to endoreduplicate (*Ullah et al., 2008*). Although endoreduplication is not expected in crypt cells, enlarged polynucleated cells have been reported to remain in the epithelium without dying in a recent light-sheet organoid imaging study tracking the progeny of a cell after cytokinesis failure induced by the inhibition of LATS1 (*de Medeiros et al., 2022*), which is phosphorylated by CDK1 during mitosis (*Furth and Aylon, 2017*). Thus, we chose to replicate this phenotype to show the capacity of our model to predict possible complex responses in the intestine. Following CDK1 inhibition, we detected oversized cells in the ABM (*Figure 4A*). The inhibition of the activation of cyclins A and B altered the modelled protein profiles, disturbing progression through G2 and M-phase and preventing the cell mass from dividing before reinitiating a new cycle (*Figure 4B*). Thus, a cell could either be (i) unaffected if it was at the

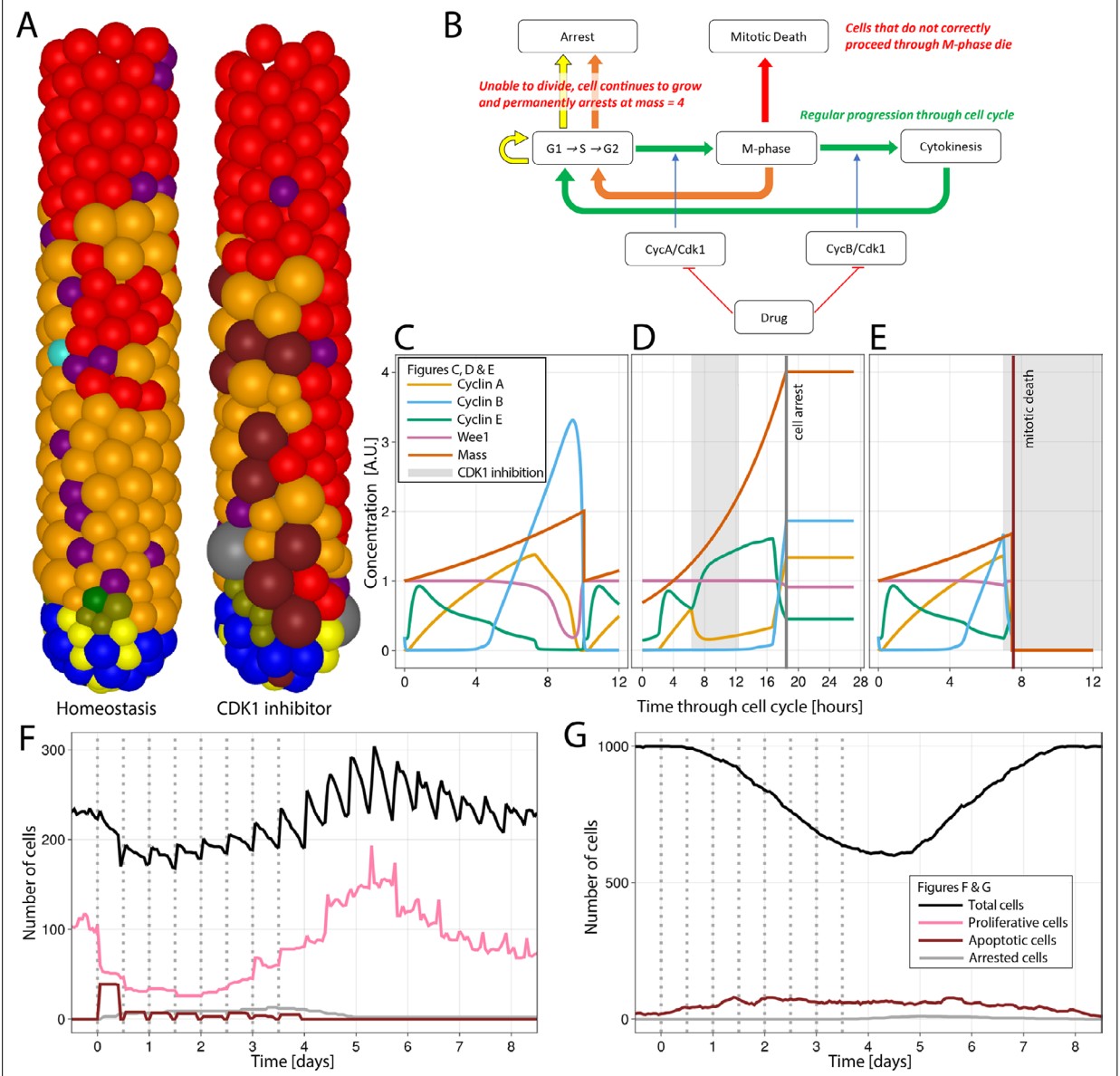

**Figure 4.** Simulation of CDK1 inhibition for 6 hr, every 12 hr for four consecutive days in the agent-based model (ABM) and impact on the cell cycle and crypt and villus organization. All cell lineages are recorded during treatment and a few days after recovery of the simulated crypt for comparison with homeostasis. (**A**) A simulated 3D image of a crypt in homeostasis (left) and a crypt subjected to CDK1 inhibition (right). Following CDK1 inhibition, the simulated crypt exhibits apoptotic cells and oversized cells unable to correctly complete the cell cycle and eventually undergo cell cycle arrest. Colour code provided here for apoptotic and arrested cells and in *Figure 1A* for the rest of cells. (**B**) Flowchart showing the regular progression through the cell cycle (green path) disturbed by CDK1 inactivation. A disorderly restart of the cycle, leading to enlarged cells, is observed when CDK1 inhibition prevents cells from entering (yellow path) or completing M-phase (orange path) by early reduction of cyclin B, with premature restart of G1 (orange path). Cells in M-phase subjected to greater reduction of cyclins A and B that completely disrupts the protein network undergo mitotic death (red path). (**C**) Cell cycle protein dynamics in homeostasis. (**D, E**) Altered cell cycle protein profile by CDK1 inhibition, resulting in premature restart of G1 and arrest of enlarged cell (**D**) and in disruption of the protein network and cell death (**E**). Protein concentrations given in arbitrary units (A.U.). Cell dynamics in simulated crypts (**F**) and villi (**G**) during CDK1 inhibition period and recovery. The dynamics of all cell lineages are reported in *Appendix 1—figure 1*. Discontinuous bars denote the beginning of CDK1 inhibition period.

early stages of the cycle (*Figure 4C*); or (ii) restart the cell cycle if CDK1 was inhibited while the cell was at the end of G2 and unable to enter M-phase or in M-phase and unable to complete cytokinesis. In this case, the inhibition of cyclins A and B led to an early increase in cyclin E and the premature restart of G1 with the generation of oversized cells, which are ultimately arrested (*Figure 4D*); or (iii) cells in M-phase can undergo mitotic death if the reduction of cyclins A and B severely disrupts the

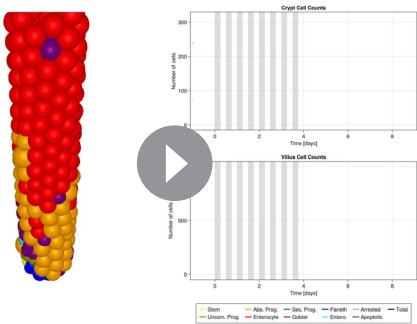

**Video 2.** Simulated cell dynamics in the epithelium subjected to CDK1 inhibition for 4 d. Plots depict changes in the number of cells in the crypt and villus during the simulation. Colour code of cell types is included below plots.

https://elifesciences.org/articles/85478/figures#video2

protein network (*Figure 4E*). Hence, the failure to culminate M-phase resulted in cell death or generation of oversized, nonproliferating cells, which led to a reduction of the crypt overall cell number (*Figure 4F*) and the turnover of villus cells (*Figure 4G*). *Appendix 1—figure 1* shows the response of all cell lineages to CDK1 inhibition, and *Video 2* shows the 3-D visualization of the crypt during this treatment.

Altogether our ABM enables the simulation of how disruptions of the cell cycle protein network span across scales to generate complex pheno-types, such as giant cells, and impact on the integrity of the crypt and villus structure.

## A practical application of the ABM to describe 5-fluorouracil (5-FU)-induced epithelial injury at multiple scales

5-FU is a well-studied and commonly administered cancer drug (*Longley et al., 2003*) with reported high incidence of gastrointestinal adverse effects in treated patients (*Stein et al., 2010*). 5-FU is a pyrimidine antimetabolite cytotoxin which has multiple mechanisms of action upon conversion to several nucleotides that induce DNA and RNA damage (*Longley et al., 2003*). Antimetabolites resemble nucleotides and nucleotide precursors that inhibit nucleotide metabolism pathways, and hence DNA synthesis, as well as impair the replication fork progression after being incorporated into the DNA (*Helleday et al., 2008*).

To explore the performance of our ABM to predict epithelial injury, we used results from experiments in mice dosed with 50 and 20 mg/kg of 5-FU every 12 hr for 4 d to achieve drug exposures similar to those observed in patients (*Jardi et al., 2023*). 5-FU pharmacokinetics is metabolized into three active metabolites FUTP, FdUMP, and FdUTP (*Longley et al., 2003*). Based on previous reports, we assumed that FUTP is incorporated into RNA of proliferative cells, leading to global changes in cell cycle proteins (*Pritchard et al., 1997*), while FdUTP is incorporated into DNA (*Longley et al., 2003*) during S-phase, resulting in the accumulation of damaged DNA. In our model, DNA and/or RNA damage can be repaired or lead to cell arrest or apoptosis (*Figure 5A*). We did not implement the inhibition of thymidylate synthase (TS) by FdUMP because the impact of this mechanism on intestinal toxicity is not completely understood (*Pritchard et al., 1997*). A previously published 5-FU PK model (*Gall et al., 2023*) was integrated into the ABM to describe the dynamic profile of the concentration of 5-FU and its metabolites in plasma and GI epithelium after dosing (*Figure 5B*).

*Figure 5C* shows the cell cycle protein dynamics and fate decision when 5-FU challenge took place at the beginning of S-phase and led to the accumulation of relatively high levels of DNA damage which triggered cell death at the G2-M-phase checkpoint. When the challenged cell was at the end of S-phase, the accumulated levels of DNA damage were not high enough to be detected at the G2-M-phase checkpoint and the cell finished the cycle and restarted a new cycle at a slower rate due to concurrent RNA damage and relatively low level of DNA damage (*Figure 5D*).

*Figure 5E* shows that predicted and observed Ki-67-positive cells declined gradually over time at all positions in the crypt during the 5-FU high-dose treatment. However, the numbers recovered, reaching values above baseline, 2 d after the interruption of 5-FU administration. The increased rebound of the proliferative crypt compartment after treatment was captured in our ABM by the implemented BMP-mediated feedback mechanism from mature enterocytes to proliferative cells (see Appendix 1, Section 1.7.4). For this treatment, both simulated and observed total number of cell,s in the crypt followed the same pattern as the proliferative compartment (*Figure 5F*), while the decline in villus cells started later and took longer to achieve full recovery (*Figure 5G*). *Appendix 1—figure 2A and B* shows the response of all cell lineages during this treatment, and *Video 3* shows the 3-D visualization of the simulated crypt and changes in signalling pathways and cell composition during

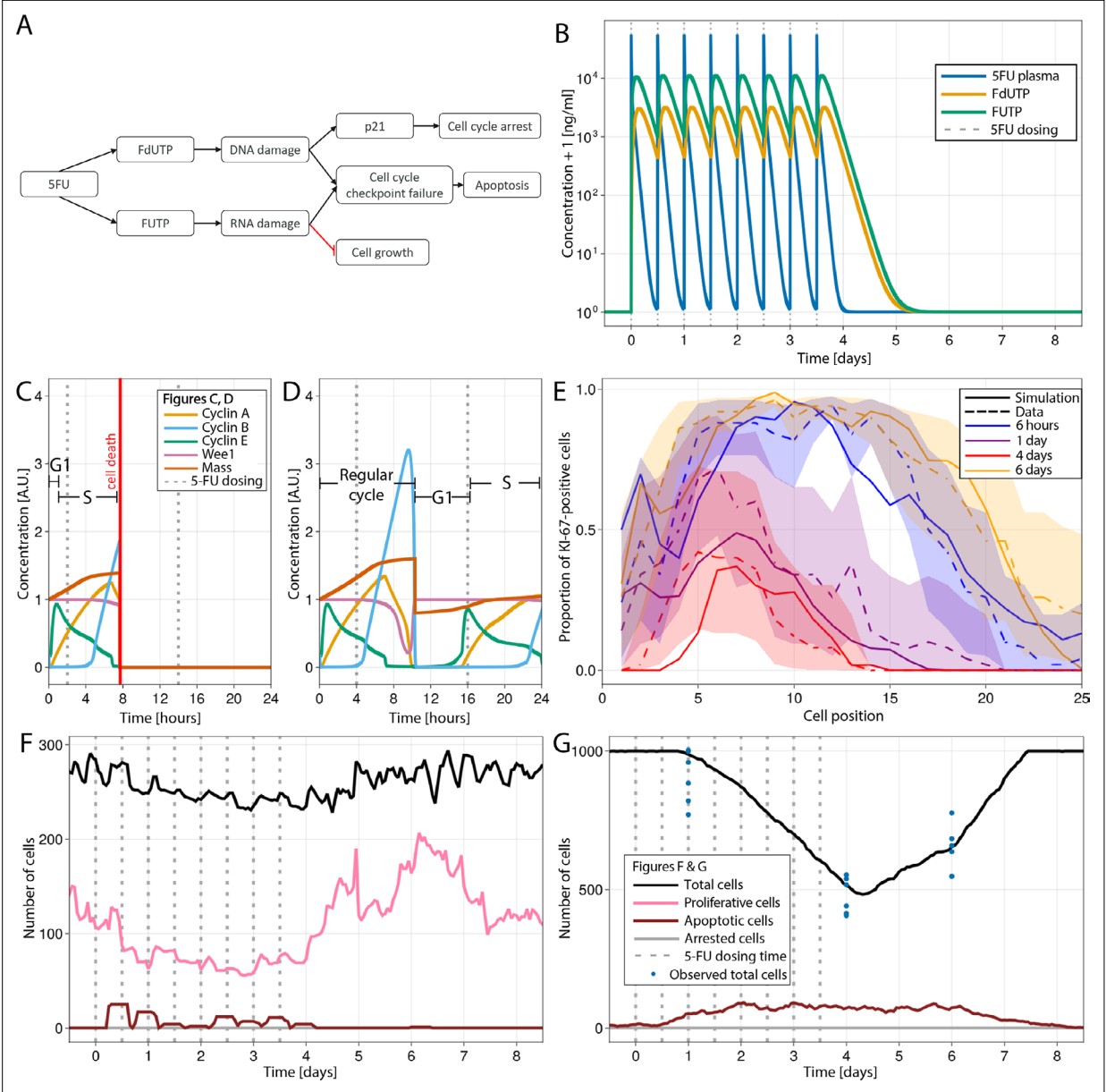

**Figure 5.** Modelling 5-fluorouracil (5-FU) (50 mg/kg twice a day for 4 d) induced injury at several scales in mouse small intestinal epithelium. (**A**) Diagram showing the implemented mechanism in the agent-based model (ABM) to describe DNA and RNA damage and cell cycle disruption driven by 5-FU metabolites. Cells trigger the apoptotic pathway if relatively high levels of RNA and/or DNA damage are detected at the cycle checkpoints. Lower levels of DNA damage induced P21 activation, which, together with RNA damage, slow down and could eventually arrest the cycle. (**B**) Predicted concentration (ng/ml) of 5-FU, FUTP, and FdUTP in plasma in mouse (pharmacokinetics model of 5-FU described in *Gall et al., 2023*). (**C, D**) Cell cycle protein dynamics and fate decision when 5-FU challenge starts (**C**) prior to or at the beginning of S-phase, leading to DNA damage and cell death at the G2-M-phase checkpoint, and (**D**) at the end of S-phase, resulting in not enough DNA damage, the cell finishes the cycle. (**E**) Predicted (solid line) and observed (dashed line) proportions of Ki-67-positive cells along the crypt axis at 6 hr, 1 d, 4 d, and 6 d during the 5-FU treatment period. Shadows depict the 95% confidence interval of our simulated staining results assuming that the proportion of staining cells has a beta distribution and estimating its error from experimental data. (**F, G**) Predicted (lines) and observed (symbols) number of cells in the crypt (**F**) and villus (**G**). Vertical bars represent dosing times. Symbols represent cell counts from individual mice.

the high-dose 5-FU challenge. The low dose of 5-FU had a minor impact on crypt proliferation and villus integrity, which was also recapitulated by the model (*Appendix 1—figure 2C–E*).

Overall, the ABM recapitulates DNA and RNA damage, resulting in cell cycle disruption associated with 5-FU administration and describes the propagation of the injury across scales to disturb epithelial integrity. The loss of epithelial barrier integrity is widely accepted to be the triggering event

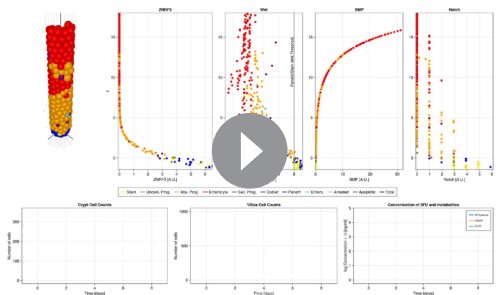

**Video 3.** Simulated cell and signalling molecular dynamics in the epithelium following the administration of 50 mg/kg of 5-fluorouracil (5-FU) twice a day for 4 d in mouse. Plots depict changes in signal abundance across the crypt longitudinal axis (z), in the number of cells in the crypt and villus, and concentration of 5-FU and metabolites during the simulation. Signals expressed in arbitrary units (A.U.). Colour code of cell types is included below plots.

https://elifesciences.org/articles/85478/figures#video3

of chemotherapy-induced diarrhoea (*McQuade et al., 2016*) which is reported in mice at the doses used in this study (*Jardi et al., 2023*) as well as observed in patients undergoing equivalent treatments (*Morawska et al., 2018*).

## Discussion

We have built a multi-scale ABM of the small intestinal crypt with self-organizing, stable behaviour that emerges from the dynamic interaction of the Wnt, Notch, BMP, and ZNRF3/RNF43 pathways orchestrating cellular fate and feedback regulatory loops and includes contact inhibition of proliferation, RNA and DNA metabolism, and the cell cycle protein interaction network regulating progression across division stages.

In our model, the stability of the niche is achieved by a negative feedback mechanism from stem cells to Wnt respondent cells that resembles the reported turnover of Wnt receptors by ZNRF3/RNF43 ligands secreted by stem cells (*Hao et al., 2012*; *Koo et al., 2012*; *Clevers, 2013b*; *Clevers and Bevins, 2013c*). Wnt signals generated from mesenchymal cells and Paneth cells at the bottom of the crypt are tethered to receptive cells and divided between daughter cells upon division, which forms a decreasing Wnt gradient towards the villi that stimulates cell proliferation and ensures stemness maintenance (*Farin et al., 2016*; *Sato et al., 2011*). The model also implements the BMP signalling counter-gradient along the crypt–villus axis by resembling the production of diffusive BMP signals by mesenchymal telocytes abundant at the villus base as well as the activity of BMP antagonist molecules secreted by trophocytes located just below crypts (*McCarthy et al., 2020*). This BMP signalling gradient forms an additional negative feedback mechanism that regulates the size of the crypt proliferative compartment and recapitulates the modulation of BMP secretion by mesenchymal cells via villus cells-derived hedgehog signalling (*Büller et al., 2012*; *van den Brink et al., 2004*).

Another novel feature of our model is the inclusion of the dynamics of the protein network governing the phases of cell division (*Csikász-Nagy et al., 2006*). Moreover, in our model, the cell cycle protein network responds to environmental mechanical cues by adapting the duration of the cycle phases. Cells in crowded environments subjected to higher mechanical pressure, such as stem cells in the niche, exhibit longer cell cycles (*Wright and Alison, 1984*; *Marshman et al., 2002*; *Potten et al., 1997*) while progenitors in the transit-amplifying compartment adapt their cell cycle protein dynamics to mainly shorten G1-phase (*Wright and Alison, 1984*; *Carroll et al., 2018*) and proliferate more rapidly. This model feature recapitulates the widely reported YAP-mediated mechanism of contact inhibition of proliferation under physical compression (*Halder et al., 2012*; *Aragona et al., 2013*; *Low et al., 2014*). Interestingly, it has been reported that stiff matrices initially enhance YAP activity and proliferation of in vitro cultured intestinal stem cells by promoting cellular tension (*Gjorevski et al., 2016*); however, that study also proposes that the resulting colony growth within a stiff confining environment may give rise to compression YAP inactivation retarding growth and morphogenesis (*Gjorevski et al., 2016*).

Furthermore, our model considers that the mechanical regulation of the cell cycle interacts with signalling pathways to maintain epithelial homeostasis, but also to trigger cell dedifferentiation if required. Cells with longer cycles accumulate more Wnt and Notch signals, leading to the maintenance of the highly dynamic niche by replacement of Paneth and stem cells. Cells located outside the niche exhibit shorter cycles and cannot effectively accumulate enough Wnt signals to dedifferentiate into stem cells in homeostatic conditions. However, in case of niche perturbation, progenitor cells reaching the niche as well as existing Paneth cells in the niche are able to dedifferentiate into stem cells after regaining enough Wnt signals, which replicates the injury recovery mechanisms observed in the crypt (*Hageman et al., 2020*; *Tetteh et al., 2016*). Our model also concurs with

experimental results suggesting that Lgr5+ stem cells are essential for intestinal homeostasis and that their persistent ablation compromises epithelial integrity (*Tan et al., 2021*).

Altogether, our model implements qualitative and quantitative behaviours to better simulate the functional heterogeneity of the intestinal epithelium at multiple scales. One of the important applications of our modelling approach lies in the development of safer oncotherapeutics. The model enables the prediction of intestinal injury associated with efficacious dosing schedules in order to minimize toxicity while maintaining the efficacy of investigational drugs. We demonstrated the application of our model to predict potential intestinal toxicity phenotypes induced by CDK1 inhibition as well as describe the disruption of the epithelium at multiple scales triggered by RNA and DNA damage, leading to the loss of integrity of the intestinal barrier and diarrhoea following 5-FU treatment. The drug-induced perturbation of other cell cycle proteins or signalling pathways, already integrated into the model, is straightforward to simulate with the current version of the model while the resolution of molecular networks can be increased, or new pathways incorporated into the ABM, to describe additional drug mechanisms of action.

While most of the crypt biology understanding integrated in our model derives from mouse epithelial studies, human-derived intestinal organoids and microphysiological systems, now routinely used in research, can provide highly precise information at the single-cell level to inform ABM development. In return, ABMs can help test hypotheses behind organoid responses in health and disease conditions. Our work highlights the importance of novel modelling strategies that are able to integrate the dynamics of processes regulating the functionality of the intestinal epithelium at multiple scales in homeostasis and following perturbations to provide unprecedented insights into the biology of the epithelium with practical application to the development of safer novel drug candidates.

## Materials and methods

### Mouse experiments

We used BrdU tracking and Ki-67 immunostaining data from previously published experiments in healthy mice (*Parker et al., 2017*; *Parker et al., 2019*) and following 5-FU treatment (*Jardi et al., 2023*). The samples from this later study (*Jardi et al., 2023*) were analysed again to count Ki-67-positive cells at each position along the longitudinal crypt axis for 30–50 individual hemi crypt units per tissue section per mouse as previously described (*Williams et al., 2016*).

### ABM development

A comprehensive description of the model can be found in Appendix 1 and *Appendix 1—table 1*. The model has been made available through BioModels (MODEL2212120002) (*Malik-Sheriff et al., 2020*)

## Acknowledgements

The authors acknowledge financial support from TransQST consortium. This project has received funding from the Innovative Medicines Initiative 2 Joint Undertaking under grant agreement no. 116030. This Joint Undertaking receives support from the European Union's Horizon 2020 research and innovation programme and EFPIA.

---

## Additional information

#### Competing interests
Louis Gall, Ambra Bianco, Holly Kimko, Carmen Pin: Employee and shareholder of AstraZeneca Plc. Ferran Jardi: Employee of Johnson & Johnson. Lieve Lammens: Employee and shareholder of Johnson & Johnson. The other authors declare that no competing interests exist.

## Funding

| Funder | Grant reference number | Author |
|---|---|---|
| European Federation of Pharmaceutical Industries and Associations | Innovative Medicines Initiative 2 No. 116030 | Louis Gall<br>Carrie Duckworth<br>Ferran Jardi<br>Lieve Lammens<br>David Mark Pritchard<br>Carmen Pin |
| Horizon 2020 Framework Programme | Innovative Medicines Initiative 2 No. 116030 | Louis Gall<br>Carrie Duckworth<br>Ferran Jardi<br>Lieve Lammens<br>David Mark Pritchard<br>Carmen Pin |

The funders had no role in study design, data collection and interpretation, or the decision to submit the work for publication.

### Author contributions

Louis Gall, Data curation, Formal analysis, Investigation, Methodology, Writing – original draft, Writing – review and editing; Carrie Duckworth, David Mark Pritchard, Supervision, Writing – review and editing; Ferran Jardi, Data curation, Writing – review and editing; Lieve Lammens, Aimee Parker, Data curation; Ambra Bianco, Holly Kimko, Supervision; Carmen Pin, Conceptualization, Resources, Supervision, Funding acquisition, Investigation, Methodology, Writing – original draft, Writing – review and editing

### Author ORCIDs

Louis Gall ⓘ https://orcid.org/0000-0002-1805-2357
David Mark Pritchard ⓘ http://orcid.org/0000-0001-7971-3561
Carmen Pin ⓘ http://orcid.org/0000-0001-8734-6167

### Ethics

All experiments were performed in an Association for Assessment and Accreditation of Laboratory Animal Care approved rodent facility and in accordance with the applicable animal welfare guidelines and legislation. Experimental procedures were approved by the institutional ethics committee. Ten-week-old male C57/BL6Y mice were obtained from Charles River (France). Mice were housed in polysulfon cages with corncob bedding under standard conditions of room temperature (21°C ± 2), relative humidity (55% ± 15) and a 12-h light cycle. Water and a certified rodent pelleted maintenance diet were supplied ad libitum. Nest material and rodent retreats were provided for environmental enrichment.

### Decision letter and Author response

Decision letter https://doi.org/10.7554/eLife.85478.sa1
Author response https://doi.org/10.7554/eLife.85478.sa2

## Additional files

### Supplementary files
• MDAR checklist
• Source code 1. Julia implementation of the agent-based model.

### Data availability

The current manuscript is a computational study. No data have been generated for this manuscript. Modelling code is uploaded as Source code 1.

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

# Appendix 1

## Technical description of the intestinal epithelial ABM

The model primarily focuses on describing the spatiotemporal dynamics of single epithelial cells, interacting physically and biochemically in the mouse intestinal crypt, undergoing division cycles or differentiating into mature epithelial cells. Single cells both generate and respond to signals and mechanical pressure in the crypt–villus geometry to generate a self-organizing tissue.

Below we describe the assumptions and hypotheses that underpin the model, regarding (1) geometry; (2) cell cycle proteins and cellular growth; (3) drug perturbation of the cell cycle proteins: Cdk1 inhibition; (4) DNA and RNA synthesis; (5) drug perturbations of RNA and DNA synthesis: 5-FU-induced RNA and DNA damage; (6) mechanical cell interactions and contact inhibition; (7) biochemical signalling; (8) cell fate: proliferation, differentiation, arrest, and apoptosis; (9) ABM simulation of Ki-67 and BrdU staining; (10) 'What-if' analysis; and (11) model implementation and parameterization.

## Geometry

To recreate the morphology of the crypt, we chose the common idealized 'test tube' crypt geometry of a hemisphere attached to a cylinder, which describes the basement membrane that the cells are attached to. The parameters describing the average morphology of the crypt, that is, the height and circumference of the 'tube', in mouse jejunum and ileum are described in *Appendix 1—table 1*.

Cells on the villus are terminally differentiated and can be assumed to migrate on a conveyor belt at constant velocity (*Parker et al., 2017*). Given these simple dynamics, to save computational power and time we modelled individual cells on the villus without spatial granularity. Cells that reach the top of the crypt are collected into a villus compartment. Shedding from the villus tip is mimicked by removing the oldest cells when the number of cells exceeds the maximum capacity of the villus, which is described in *Appendix 1—table 1*. Cells on the villus keep all properties and still age and undergo apoptosis if required, though in homeostatic conditions cells are usually shed into the lumen before becoming senescent.

**Appendix 1—table 1.** Parameter values of the agent-based model (ABM) model of the mouse small intestinal crypt.

| Parameter | Value | | Source/justification |
|---|---|---|---|
| | Jejunum | Ileum | |
| **Geometry** | | | |
| $C_h$, maximum fold-increase of the height of the crypt | 1.2 | | Calibration of ABM to experimental data in drug-injured crypt |
| $q$, Hill coefficient of height scaling | 3.5 | | Calibration of ABM to experimental data in drug-injured crypt |
| Crypt circumference | 9 distance units* | | Calibration of ABM to experimental data in a homeostatic crypt |
| | | | Due to cell compression, this results in a crypt circumference of 10 cells |
| Villus length | 1000 cells (700 for cell ablation) | | |
| **Biochemical signalling** | | | |
| *BMP/Indian hedgehog feedback* | | | |
| $z_0$, z coordinate at the top of the crypt in homeostatic conditions (measured from the edge of stem cell niche in the direction of the longitudinal crypt–villus z-axis) | 18 distance units*. | 16 distance units*. | Calibration of ABM to experimental data in homeostatic and drug-injured crypt |
| | | | Due to cell compression, this results in a homeostatic crypt height of 22–26 (jejunum) and 20–24 (ileum) measured in cell positions |

*Appendix 1—table 1 Continued on next page*

*Appendix 1—table 1 Continued*

| Parameter | Value | | Source/justification |
|---|---|---|---|
| | Jejunum | Ileum | |
| $z_{50}$, z coordinate at which the number of mature enterocytes becomes greater than the number of absorptive progenitors in homeostatic conditions (measured from the edge of stem cell niche in the direction of the longitudinal crypt–villus z-axis) | 14.5 distance units*. | 13 distance units*. | Calibration of ABM to experimental data in a homeostatic crypt<br><br>Due to cell compression, this corresponds to cell position 18 (jejunum) and 16 (ileum) |
| $E_h$, homeostatic enterocyte count | 940 cells | 920 cells (644 for cell ablation experiment) | Calibration of ABM to experimental data in homeostatic and drug-injured crypt (*Umar, 2010*)<br><br>Cell ablation experiment scaled using 0.7× reduction in villus length |
| $p$, Hill coefficient of Hedgehog/BMP feedback | 8.5 | 4 | Calibration of ABM to experimental data in a drug-injured crypt |
| $A$, level of BMP signals at height $z_{top}$ | 64 | | Calibration of ABM to experimental data in homeostatic crypt |
| $B$, exponential transformation of the diffusion coefficient of BMP signals | 8 | | Calibration of ABM to experimental data in homeostatic crypt |
| *Wnt signalling* | | | |
| $k_{Wnt}$ | 152.80 | | Calibration of ABM to experimental data in homeostatic crypt |
| *WntRange*, Wnt signalling short-range field around Paneth cells and Wnt-emitting mesenchymal cells where Wnt signals tethered to receptive cells | 0.35 distance units* | | Calibration of ABM to experimental data in homeostatic crypt |
| $Mesenchymal_{niche}$, number of Wnt emitting mesenchymal cells surrounding the stem cell niche | 32 cells | | Assumed equal to the total number of epithelial cells in the niche in homeostatic conditions (*Wright and Alison, 1984*; *Snippert et al., 2010*) |
| $M_{Wnt}$, maximum number of Wnt signals a cell can have tethered | 128 | | Calibration of ABM to experimental data in homeostatic crypt<br><br>Value chosen to be a power of 2 to facilitate dividing Wnt signals in half upon cellular division |
| *Notch signalling* | | | |
| $k_{notch}$ | 200 | | Calibration of ABM to experimental data in homeostatic crypt |
| Notch range | 0 during homeostasis, increasing to one cell unit with drop in local cell density | | Calibration of ABM to experimental data in homeostatic and drug-injured crypt |
| *ZNRF3/RNF43 signalling* | | | |
| $Z$, maximum signal strength immediately around the emitting cell | 1 | | Calibration of ABM to experimental data in homeostatic crypt |
| $L_{ZNRF3}$, length scale of ZNRF3 signalling. | 1 distance units* | | Calibration of ABM to experimental data in homeostatic crypt |
| $K$, regulates the dependence of the decay rate of cell surface-tethered Wnt molecules on ZNRF3/RNF43 signals | 19.1 | | Calibration of ABM to experimental data in homeostatic crypt |
| $ZNRF3$, ZNRF3 signal strength experienced at the edge of niche | 3.5 | | Calibration of ABM to experimental data in homeostatic crypt. Maintains homeostatic number of stem cells, ≈14-16 (*Snippert et al., 2010*) |

*Appendix 1—table 1 Continued on next page*

*Appendix 1—table 1 Continued*

| Parameter | Value | | Source/justification |
|---|---|---|---|
| | Jejunum | Ileum | |
| $u$, Hill coefficient of ZNRF3 feedback. | 1 | | Calibration of ABM to experimental data in a drug-injured crypt |
| **Contact inhibition and mechanical parameters** | | | |
| $p_0$ | 3.2 | | Calibration of ABM to experimental data in homeostatic crypt |
| $\epsilon_{Paneth-Paneth}$ , adhesive constant for Paneth–Paneth interactions | 0.1111 | | Calibration of ABM to experimental data in homeostatic crypt |
| $\epsilon$, adhesive constant for all other interactions | 0.01111 | | Calibration of ABM to experimental data in homeostatic crypt |
| $\nu$, Poisson ratio | 0.4 | | Based on published data (*Geissler and Hecht, 1981*; *Mokbel et al., 2020*; *Mahaffy et al., 2004*) |
| $E$, Young's modulus | 25 kPa (Paneth), 4 kPa (all others) | | *Pin et al., 2015* |
| $\mu_{fric.}$ | $\frac{\pi\left(r^2 - p_{ij}^2\right)}{10000}$ | | Calibration of ABM to experimental data in homeostatic crypt<br><br>Maintains published cell transfer velocity of 1 cell/hr (*Potten, 1998*) |
| $\mu$ | $\frac{1+2\pi r_{cell}^2}{3000}$<br>(multiplied by 10,000 for Paneth cells) | | Calibration of ABM to experimental data in homeostatic crypt<br><br>Maintains published cell transfer velocity of 1 cell/hr (*Potten, 1998*) |
| **Cell timescales** | | | |
| $t_{cycle}^{short}$ | 10 hr | | *Wright and Alison, 1984* |
| $t_{cycle}^{long}$ | 21.5 hr | | *Schepers et al., 2011* |
| Paneth cell lifespan | 54 d | | *Ireland et al., 2005* |
| Other cell lifespan | 6 d (jejunum), 6.5 d (ileum)<br>(4 d for villus cells in Barker experiment) | | Calibration of ABM to experimental data in drug-injured crypt |
| Apoptosis duration | 12 hr | | *Sundquist et al., 2006* |
| Paneth ->Stem de-differentiation duration | 48 hr (reversible for first 36 hr) | | Calibration of ABM to experimental data in homeostatic crypt (*Tan et al., 2021*) |
| Absorptive progenitor ->Enterocyte differentiation duration | 0 hr | | Calibration of ABM to experimental data in homeostatic crypt |
| All other differentiations duration | 4 hr | | *Stamataki et al., 2011* |
| Duration of Ki-67 positivity | 6 hours post-differentiation.<br>6/8/10/12 hr after drug-induced cell cycle interruption in G1/S/G2/M-phase | | Calibration of ABM to experimental data in homeostatic and drug-injured crypt.(*Miller et al., 2018*) |
| $K_{BrdU}$, theoretical maximum level of BrdU | 5.7 | | Calibration of ABM to experimental data in homeostatic crypt |
| **Cell fate decision parameters** | | | |
| Stem ->Paneth Notch threshold | 3 | | Calibration of ABM to experimental data in homeostatic crypt |
| Absorptive/secretory Notch threshold | 2 | | Calibration of ABM to experimental data in homeostatic and drug-injured crypt |

*Appendix 1—table 1 Continued on next page*

*Appendix 1—table 1 Continued*

| Parameter | | Value | | Source/justification |
|---|---|---|---|---|
| | | Jejunum | Ileum | |
| Paneth ->Stem emergency Notch threshold | | 5 | | Calibration of ABM to experimental data in homeostatic crypt. Qualitatively reproduces (*Yu et al., 2018*) |
| Paneth/stem Wnt threshold | | 64 | | Arbitrary parameterization |
| **Cell cycle modifications** | | | | |
| $c_{mass}$ | $c_{mass}^{short}$ | 0.962 | | Calibrated to enable variable division time responding to mechanical cues |
| | $c_{mass}^{long}$ | 1.175 | | |
| $c_{Vsi}$ | $c_{Vsi}^{short}$ | 0.1 | | Calibrated to modify G1-phase duration for variable division time |
| | $c_{Vsi}^{long}$ | 1 | | |
| $KA_{Wee1p}$ | $KA_{Wee1p}^{short}$ | 2.76 | | Calibrated to maintain S-phase duration for variable division time |
| | $KA_{Wee1p}^{long}$ | 0.97 | | |
| **DNA/RNA module** | | | | |
| $v_1$ | | 2 | | Calibration of ABM to experimental data in homeostatic crypt |
| $v_2$ | | 5 | | Calibration of ABM to experimental data in drug-injured crypt |
| $s$ | | 3 | | Calibration of ABM to experimental data in drug-injured crypt |
| $K_{Vsi}$ | | 5 | | Calibration of ABM to experimental data in drug-injured crypt |
| $t$ | | 2 | | Calibration of ABM to experimental data in drug-injured crypt |
| **Drug simulation parameters** | | | | |
| *Cdk1 inhibitor* | | | | |
| $k_{drug,CycA}$ | | 70 | | N/A |
| $k_{drug,CycB}$ | | 140 | | N/A |
| Mass threshold | | 4 | | Mass of cell determining cycle arrest for cells unable to undergo mitosis |
| *5-FU simulation* | | | | |
| $d_{DNA}$ | | 50.8 | | Calibration of ABM to experimental data in drug-injured crypt |
| $K_{FdUTP}$ | | 13100 | | Calibration of ABM to experimental data in drug-injured crypt |
| $n$ | | 5.82 | | Calibration of ABM to experimental data in drug-injured crypt |

*Appendix 1—table 1 Continued on next page*

*Appendix 1—table 1 Continued*

| Parameter | Value | | |
|---|---|---|---|
| | Jejunum | Ileum | Source/justification |
| $d_{RNA}$ | 30.3 | | Calibration of ABM to experimental data in drug-injured crypt |
| $K_{FUTP}$ | 1610 | | Calibration of ABM to experimental data in drug-injured crypt |
| $m$ | 1.940 | | Calibration of ABM to experimental data in drug-injured crypt |
| Computational parameters | | | |
| $dt$, timestep for movement | 0.0001 d | | N/A |
| $dt_{cycle}$, timestep for cell cycle | 0.00001 d | | N/A |

*All distances are normalized such that an average, isolated cell has a diameter of one distance unit. We have assumed cells are deformable and hence lose the spherical shape upon compression so that the cell diameters, in both the *z*-axis direction (longitudinal crypt–villus axis), as well as in the *y*–*x* plane (crypt transversal circumference), are smaller than one unit and result in inequality between the number of cells and the distance units.

## Cell cycle proteins and cellular growth

The division cycle of cells is controlled by a network of interacting proteins which include cyclins, cyclin-dependent kinases (CDKs), and a suite of ancillary proteins (*Morgan and Morgan, 2007*). The discrete events of the cell cycle, such as DNA replication in S-phase and the various stages of mitosis, are regulated by the activity of this protein network, whose components go through a careful, conserved series of peaks and troughs at the correct pace to complete all processes of the cycle. The dynamics of this protein interaction network is simulated in each cell of the ABM and controls cell division and differentiation.

We have used the model of *Csikász-Nagy et al., 2006* that recreates the mammalian cell cycle and is available in Biomodels (*Le Novère and Csikasz-Nagy, 2006*). This model is an extension of the pioneering work of Novak and Tyson that helped reveal the complex nonlinear dynamics of the cell cycle proteins (*Novak and Tyson, 1993*; *Novak et al., 2001*; *Novák and Tyson, 2004*). The Csikasz-Nagy model provides multiple necessary features such as core cell cycle proteins, a mass variable that can be coupled to the volume of the single cells in our ABM and sufficient mechanistic detail to enable a detailed description of drug–cycle interactions. The model compromises 14 variables that describe the dynamics of the concentration of the main cell cycle proteins as oscillations between alternating peaks and troughs. G1-phase is the default opening state, with low levels of cyclins A, B, and E and high level of Wee1. The level of cyclin D grows exponentially throughout the cycle and is halved between daughter cells after mitosis. S-phase begins with the increase in cyclin E and ends when Wee1 drops to reach its trough. G2-phase is characterized by low Wee1 and high cyclin A, ending with the drop of cyclin A. M-phase ends when cyclin B falls and the cell divides and restarts the cycle in G1.

Stem cells have been reported to have a longer division cycle than absorptive progenitor cells (*Wright and Alison, 1984*; *Marshman et al., 2002*; *Potten et al., 1997*). We hypothesize that this is due to contact inhibition mechanisms caused by increased intercellular forces in the crowded, constrained niche. This implies that the duration of the cycle may significantly vary among single cells. To implement cycles of varying duration in our ABM, we describe below a series of required adjustments in the Csikasz-Nagy model that basically involve changes in the duration of the full cycle, the re-adjustment of the length of the cycle phases, primarily G1 and S-phase, and the modulation of the dynamics of the model mass variable.

To change the duration of the cell cycle, $t_{cycle}$ , we rescaled the time coordinate: $t \to \frac{\tau}{t_{cycle}}t$, where $\tau = 140.027$ h is the original period of the model (*Csikász-Nagy et al., 2006*) and $t_{cycle}$ is determined by the internal pressure of the cell as detailed below in Section 1.6.

Without further modifications of the Csikasz-Nagy model (*Csikász-Nagy et al., 2006*), the duration of all cycle phases would be scaled in proportion with changes in $t_{cycle}$. However, not all phases are proportionally shortened in fast cycling healthy cells (*Csikász-Nagy et al., 2006*). The

shorter cycle duration in absorptive progenitors is likely due to shortening/omission of G1-phase as reported for rapid cycling progenitors (**Wright and Alison, 1984**; **Carroll et al., 2018**), while the duration of S-phase is less variable (**Wright and Alison, 1984**) with reported values of 8 hr for mouse ileal epithelium (**Wright and Alison, 1984**).

Regarding G1-phase, the p27 protein has been reported to regulate the duration of G1 by preventing the activation of cyclin E-Cdk2 which induces DNA replication and defines the beginning of S-phase (**Morgan and Morgan, 2007**). We hypothesized that fast cycling cells have low levels of p27 which results in earlier DNA replication, bringing forward the start of S-phase and shortening the length of G1. In support of this hypothesis, it has been experimentally demonstrated that inhibiting p27 has no effect on the proliferation of absorptive progenitors (**Zheng et al., 2008**). In the Csikasz-Nagy model (**Csikász-Nagy et al., 2006**), the duration of G1 can be modulated through the parameter $V_{si}$, which is the basal production rate of p21/p27 (in the Csikasz-Nagy model, the p21 and p27 proteins are represented by a single variable, here we refer to that model quantity as p21/p27).

Additionally, the end of S-phase is associated with the decrease in Wee1 to basal levels due to Cdc14-mediated phosphorylation of Wee1. In the Csikasz-Nagy model (**Csikász-Nagy et al., 2006**), this reaction is described by a Goldbeter–Koshland function, which includes the parameter $KA_{Wee1p}$ to regulate the level of Cdc14 required for the phosphorylation of Wee1.

Therefore, we modified these two parameters, $V_{si}$ and $KA_{Wee1p}$, to ensure that variations of the cycle duration mostly impact on G1 while the length of S-phase remains constant. We assumed that the value of the two parameters scales linearly with the duration of the division cycle, $t_{cycle}$, between a lower and upper bound, which prevent aberrant behaviour of the cell cycle model in the dynamically changing conditions of the crypt.

$V_{si}$ is scaled according to

$$V_{si} \rightarrow C_{Vsi} V_{si}, \qquad C_{Vsi} = \begin{cases} C_{Vsi}^{short} & t_{cycle} < t_{cycle}^{short}, \\ \dfrac{\left(C_{Vsi}^{long} - C_{Vsi}^{short}\right)\left(t_{cycle} - t_{cycle}^{short}\right)}{\left(t_{cycle}^{long} - t_{cycle}^{short}\right)} + C_{Vsi}^{short} & t_{cycle}^{short} \leq t_{cycle} \leq t_{cycle}^{long}, \\ C_{Vsi}^{long} & t_{cycle} > t_{cycle}^{long}, \end{cases}$$

where $t_{cycle}^{short}$ and $t_{cycle}^{long}$ denote the average duration of the cycle of fast cycling progenitors and of the slower cycling stem cells, respectively. $C_{Vsi}^{short}$ and $C_{Vsi}^{long}$ are values calibrated to ensure the correct duration of G1 for the short and long cycles, respectively, and can be found in **Appendix 1—table 1**.

Similarly, we scale $KA_{Wee1}$ using the function:

$$KA_{Wee1p} = \begin{cases} KA_{Wee1p}^{short} & t_{cycle} < t_{cycle}^{short}, \\ \dfrac{\left(KA_{Wee1p}^{long} - KA_{Wee1p}^{short}\right)\left(t_{cycle} - t_{cycle}^{short}\right)}{\left(t_{cycle}^{long} - t_{cycle}^{short}\right)} + KA_{Wee1p}^{short} & t_{cycle}^{short} \leq t_{cycle} \leq t_{cycle}^{long}, \\ KA_{Wee1p}^{long} & t_{cycle} > t_{cycle}^{long}. \end{cases}$$

Here, $KA_{Wee1}^{short}$ and $KA_{Wee1}^{long}$ are the values required to maintain constant duration of S-phase in fast and slow cycling cells and can be found in **Appendix 1—table 1**.

A further refinement required to modify the length of the cycle in the Csikasz-Nagy model comprises the mass variable. This variable doubles its value over the course of a cycle and drives the progression of the cell cycle by changing the production rates of the cycle proteins. The changing production rates affect the balance of the proteins and the duration of the cell cycle phases, which start and end at particular mass values determined by the abovementioned two rates and other parameters in the model. After the mass doubles, mitosis occurs and the mass is halved to its initial value, returning the model to the original state. From here the mass begins to grow again, repeating the cell cycle. The mass of a cell effectively tracks the cell's progress through the cell cycle.

In our ABM, $t_{cycle}$ changes continuously in each cell and modifies $V_{si}$ and $KA_{Wee1p}$ as described above, which in turn changes the mass values of the start/end of the cell cycle phases. Without

further changes in the model, this would cause the cells to not progress through the cell cycle correctly, with unbalanced phases duration and dividing at unwanted mass values, causing erroneous and unrealistic behaviour in the ABM.

This can be solved by normalizing the mass in the cell cycle model, chosen such that a cell begins at $mass = mass_{init} \approx 1$ and always divides at $mass = 2$. To do this, we first define a normalized mass variable, assumed to be proportional to the volume of the cell:

$$mass \propto V \propto r^3 \implies mass = 2\frac{r^3}{r_{final}^3},$$

where $r$ is the cell radius that takes values between $r_{init.}$ and $r_{final}$. When a proliferative cell is created, it is assigned a desired final size, $r_{final} = \sqrt[3]{2}r_*$ , where $r_* \sim N\left(0.35, 0.00875\right)$ for stem cells and $r_* \sim N\left(0.5, 0.0125\right)$ for all other cells. The mean values, 0.5 and 0.35, of the radius of progenitor and stem cells, respectively, were determined for an average, non-proliferative or proliferative progenitor cell to have, without loss of generality, a diameter of 1 while the diameter of an average stem cell is slightly smaller, 0.7. In this way, the model captures the smaller size described for columnar LGR5+ stem cells (*Barker et al., 2008*), which additionally helps recapitulate the mechanics and cell composition of the niche. The variance of the radius was determined by our implementation of the cell cycle model in the ABM. In our model, the volume of the cell is equated to the cell's mass parameter of the Csikasz-Nagy model and, hence, the cell final radius determines the duration of the cell cycle as described above. By simulating the cell cycle model, we observed that large values of the standard deviation resulted in some cells progressing through the cycle too quickly and, therefore, failing to complete the cell cycle correctly. This analysis provided an upper limit to the coefficient of variation (CV) = 0.025 to ensure all cells progress regularly through the cycle during homeostasis. This results in values of the standard deviation of the radius of 0.0125 and 0.00875 for progenitor cells and stem cells, respectively. Of note, a cell radius CV of 0.025 corresponds to a cell volume CV of about 0.075 which is not far from the reported experimental CV for cell volume, about 0.11 (*Bell and Anderson, 1967*).

We then introduce a factor $c_{mass}$ onto the four terms involving the mass variable in the cell cycle model. These terms are the basal production rates of the four cyclins A, B, D, and E, called $V_{sa}$, $V_{sb}$ , $CycD_0$, and $V_{se}$, respectively. $c_{mass}$ is given by

$$c_{mass} = \begin{cases} c_{mass}^{short} & t_{cycle} < t_{cycle}^{short}, \\ \dfrac{\left(c_{mass}^{long} - c_{mass}^{short}\right)\left(t_{cycle} - t_{cycle}^{short}\right)}{\left(t_{cycle}^{long} - t_{cycle}^{short}\right)} + c_{mass}^{short} & t_{cycle}^{short} \leq t_{cycle} \leq t_{cycle}^{long}, \\ c_{mass}^{long} & t_{cycle} > t_{cycle}^{long}. \end{cases}$$

The values $c_{mass}^{short}$ and $c_{mass}^{long}$ are values found by calibration of the cell cycle model to guarantee the cell always divides at $mass = 2$ for the short and long cycle durations.

Moreover, the cell mass is assumed to grow exponentially. A proliferative cell always reaches a final value of $mass = 2$, corresponding to the radius $r_{final} = \sqrt[3]{2}r_*$ , during the cycle time, $t_{cycle}$, so that mass must grow as

$$\frac{dmass}{dt} = \frac{ln\left(\dfrac{2}{mass_{init}}\right)}{t_{cycle}}mass.$$

This corresponds to a radial growth rate of

$$\frac{dr}{dt} = \frac{ln\left(\dfrac{r_{final}}{r_{init}}\right)}{t_{cycle}}r.$$

As $t_{cycle}$ changes dynamically through the cell cycle, the growth rate holds only for the instantaneous conditions the cell is experiencing and changes dynamically through the cell's lifetime. However, in a healthy crypt, extracellular conditions vary slowly, and the value of $t_{cycle}$ and all derived adjustment factors remain relatively unchanged.

We assumed that cells divide symmetrically. Each daughter cell has a starting radius of $r_{init.} = r_*^{parent}$ and is assigned with a new randomly generated $r_*$ value which determines $r_{final} = \sqrt[3]{2}r_*$. If $r_{init.} > r_{final}$, then we set $r_{final} = r_{init.}$ to prevent values of $mass > 2$. Since cells have a variable maximum size uncorrelated to their birth size, that is, $r_*^{parent} \neq r_*^{child}$, the initial mass value is not necessarily 1. Longer or shorter G1 phases emerge from the model to adjust the cycle duration in cells that begin with $mass < 1$ or $mass > 1$, respectively.

Proliferative daughter cells continue through its own cell cycle and proceed to grow to its own $r_{final} = \sqrt[3]{2}r_*$. Non-proliferative secretory cells differentiate from stem cells, which are smaller than other cells. To compensate for this, secretory cells grow to reach a radius $r_*$, generated as $r_* \sim N\left(0.5, 0.0125\right)$, in a time equal to $t_{cycle}^{short}$. The other type of non-proliferative cells, enterocytes, derive from absorptive progenitors and remain at $r_{init.} \approx 0.5$ without increasing size.

These definitions of mass, cell radius, and cell growth were chosen to ensure that cells have a consistent radius and guarantee that the cell cycle model correctly proceeds through all phases in each cell. Due to the varying cycle duration and extracellular conditions, this control is essential to the correct functioning of the cell cycle and overall behaviour of the ABM.

## Drug perturbations of the cell cycle model: CDK1 inhibition

We have used the Csikasz-Nagy cell cycle model to implement drug-induced perturbations of the cell cycle proteins, which are common mechanisms of action of oncotherapeutics, in our ABM. For an arbitrary component of the cell cycle model, $X$, we introduce a term dependent on the drug and $X$:

$$\frac{dX}{dt} \subset -f\left(Drug, X\right),$$

where $\subset$ means 'contains the term' and $Drug$ represents the cell concentration of the active compound/metabolite which is often described by a pharmacokinetics model. $f\left(Drug, X\right)$, quantifies the effect of the drug on $X$. This function can take several forms such as a mass-action term or a Michaelis-Menten or Hill equation. Multiple terms like this can be added concurrently to the proteins described by the Csikasz-Nagy model.

As an example, we have modelled the effects of a Cdk1 inhibition at the single-cell level in our ABM. Cdk1 binding is reported to induce nuclear translocation of cyclins A and B require to initiate mitosis (*Pesin and Orr-Weaver, 2008*). Accordingly, we have added a mass-action term onto the rate of change of the CycA/Cdk1/2 and CycB/Cdk1 complexes as follows:

$$\frac{dCycA}{dt} = Vsa + \left(Vdi + kdissa\right) \cdot TriA - kassa \cdot p27 \cdot CycA - Vda \cdot CycA - k_{drug,CycA} \cdot \left[Drug\right] \cdot CycA,$$

$$\frac{dCycB}{dt} = Vsb + \left(Vdi + kdissb\right) \cdot BCKI + V_{25} \cdot pB - kassb \cdot p27 \cdot CycB - \left(Vdb + V_{Wee}\right) \cdot CycB$$
$$- k_{drug,CycB} \cdot \left[Drug\right] \cdot CycB,$$

where $CycA$ and $CycB$ are used to refer to CycA/Cdk1/2 and CycB/Cdk1 to improve readability of the equation. $k_{drug,CycA}$ and $k_{drug,CycB}$ are parameters that quantify the drug effect, with values specified in *Appendix 1—table 1*, and $[Drug]$ denotes a theoretical drug dynamical concentration. For the simulation in *Figure 4*, we considered a CDK1 inhibitor that was administered every 12 hr for 4 d, with active cytotoxic effects for 6 hr. To model this, $[Drug]$ is given by the formula

$$\left[Drug\right] = \begin{cases} 1 & t_{dose} \leq time < t_{dose} + 6, \\ 0 & \text{otherwise}, \end{cases}$$

where $t_{dose} \in \left\{0, 12, 24, 36, 48, 60, 72, 84\right\}$ hr. Also, we considered a smaller value for $k_{drug,CycA}$ than for $k_{drug,CycB}$ to reflect the fact that CycA represents both CycA/Cdk1 and CycA/Cdk2 and only CycA/Cdk1 is inhibited.

These perturbations of the cell cycle proteins can cause incorrect progression through the cell cycle, whereupon a cell is permanently arrested. A disorderly restart of the cycle, leading to enlarged cells, is observed when CDK1 inhibition prevents cells at the end of G2 from entering M-phase or induces early reduction of cyclins A and B during M-phase, with cells failing to complete cytokinesis

and prematurely restarting G1. Cells in M-phase subjected to greater reductions of cyclins A and B, which completely disrupt the protein network, undergo mitotic death.

## DNA and RNA synthesis

Since one of the most common means of targeting the cell cycle is to exploit the effect of DNA-damaging drugs (*Helleday et al., 2008*), we added the dynamics of DNA replication during S-phase and RNA synthesis during the cell cycle.

Replicating DNA is represented by two variables, $DNA_1$ and $DNA_2$, which denote two DNA double helices formed during S-phase. $DNA_i$ is an abstraction of the proportion of undamaged DNA, which takes values from 0, representing total DNA disruption, to 1 for the whole undamaged double helix.

At the onset of S-phase, the original DNA double helix, $DNA_1$, unwinds to start the replication of strands and rapidly generates two complete sets of DNA, $DNA_1$ and $DNA_2$. This is represented in the model by

$$\{DNA_1, DNA_2\} = \{C, 0\} \xrightarrow[\text{S-phase onset}]{} \{C/2, C/2\},$$

Both $DNA_1$ and $DNA_2$ aim to reach $DNA_i = 1$: DNA synthesis is assumed to be at a faster rate during S-phase, and outside S-phase DNA synthesis takes place solely for repair at a slower rate. Hence, in healthy cells, these variables obey the following equations and algorithm:

$$\frac{dDNA_i}{dt} = k_{DNA}, \qquad k_{DNA} = \begin{cases} 0 & \text{if } DNA_i = 1, \\ v_1 & \text{if in S-phase and } 0 < DNA_i < 1, \\ \dfrac{v_1}{2} & \text{if } 0 < DNA_i < 1. \end{cases}$$

The DNA replication rate, $\nu_1$, is sufficiently fast to ensure $DNA_i$ reaches 1 during S-phase in healthy cells. Outside of S-phase, we assumed a twofold slower rate for DNA repair when the cell is not actively replicating its DNA. Values are specified in *Appendix 1—table 1*.

When the cell divides, the daughter cells are given one DNA double helix each (which are both assigned to $DNA_1$ in the respective daughter cell) to restart the cycle.

RNA levels are represented by a single $RNA$ variable. Similarly, this variable is an abstraction of the proportion of undamaged RNA in the cell, with $RNA = 1$ in a healthy cell and $RNA = 0$ for total RNA disruption. RNA synthesis is assumed to be governed by a simple linear-growth differential equation until its maximum value, $RNA = 1$, and remains at this value unless damage is induced as follows:

$$\frac{dRNA}{dt} = k_{RNA}, \qquad k_{RNA} = 0 \;\; \text{if } RNA = 1, \quad \text{else} \quad k_{RNA} = v_2,$$

with parameter values specified in *Appendix 1—table 1*.

Along with these equations for DNA and RNA levels, we added DNA and RNA-damage checkpoints to modulate the response of the Csikasz-Nagy cell cycle model to perturbations. We considered both the G1/S and the G2/M checkpoints (*Morgan and Morgan, 2007*), with cells checking their DNA and RNA levels as they progress from G1 to S-phase, and from G2 to M-phase. If the DNA and/or RNA levels are below the threshold values (see *Appendix 1—table 1*), the cell undergoes apoptosis. Checkpoint failures can occur upon drug-induced DNA or RNA damage, as explained below.

## Drug perturbations of RNA and DNA synthesis: 5-FU-induced RNA and DNA damage

Similar to the cell cycle model, drug effects are represented by adding a negative term to these differential equations:

$$\frac{dDNA_i}{dt} = k_{DNA} - f\left(Drug, DNA_i\right),$$

$$\frac{dRNA}{dt} = k_{RNA} - f(Drug, RNA),$$

where $f(Drug, X)$ could be a mass action, Hill equation, or Michaelis–Menten term quantifying the drug-induced RNA or DNA damage.

DNA damage induces increased p21 expression in cells, which prevents progression through the cell cycle and can lead to cell cycle arrest or apoptosis (*Abbas and Dutta, 2009*). To replicate this, we further modified the p21/p27 term in the Csikasz-Nagy model to respond to the DNA levels of the cell. Recall that *Vsi* was the production rate of p21/p27 in the model, and we multiplied this by $C_{Vsi}$ to moderate the production of p21/p27 (see details in 'Cell cycle proteins and cell size/growth' above). Recall that $V_{si} \rightarrow C_{Vsi}V_{si}$ ; to replicate DNA-damaged induced production of p21, we replace $C_{Vsi}$ with a bounded function dependent on the cell's DNA levels

$$C_{Vsi} \rightarrow \frac{K_{Vsi}}{1 + \left(\frac{K_{Vsi}}{C_{Vsi}} - 1\right) DNA_*{}^s}.$$

$DNA_* = DNA_1$ in G1, and $DNA_* = min(DNA_1 + DNA_2, 1)$ in all other phases and $s$ is a scaling coefficient. In homeostasis, with $DNA_1 + DNA_2 \geq 1$, this function is equal to $C_{Vsi}$ and the cell cycle model proceeds as before. With severe DNA damage, $DNA \ll 1$, the function is approximately equal to $K_{Vsi}$, always $>C_{Vsi}$, that represents the maximum fold increase in the production rate of p21, that is, $V_{si} \rightarrow K_{Vsi}V_{si}$. Parameter values can be found in *Appendix 1—table 1*. When DNA levels are reduced by drug-induced injury, this new function increases the production rate of p21/p27 which slows down the production of cyclins and the progression of the cell cycle, recapitulating a reversible cell cycle arrest for low-to-moderate DNA damage (*Shaltiel et al., 2015*).

Cell growth is dependent on the correct translation of mRNA into proteins. We hypothesized that RNA damage reduces a cell's capability of biosynthesis and leads to slower cellular growth (*Wurtmann and Wolin, 2009*). This is modelled by adding an RNA-dependent factor to the growth rate of cells:

$$\frac{dmass}{dt} = \frac{2RNA^t}{RNA^t + 1} \frac{ln\left(\frac{2}{mass_{init}}\right)}{t_{cycle}} mass,$$

where *RNA* takes as defined above values between 0 and 1 and $t$ is a scaling coefficient. Parameter values can be found in *Appendix 1—table 1*. By linking RNA integrity to cellular growth, we allow RNA damage to induce a form of cell cycle arrest, as previously reported (*Chernova et al., 1995*; *Bellacosa and Moss, 2003*).

The result of these responses to DNA and RNA damage, in combination with the cell cycle checkpoints, allows the cells in our model to exhibit a progression of responses to increasingly severe DNA and RNA damage. Cells with slightly damaged DNA and/or RNA levels grow and proliferate slowly due to impediment of their cell cycle and/or cellular growth. With moderate DNA and RNA damage, a cell enters an impermanent, reversible cell cycle arrest (characterized by a near-zero growth rate and p21-induced halt of the cell cycle). Upon interruption of the drug-induced insult, these cells will re-enter the cell cycle. In case of severe DNA and/or RNA damage, a cell will undergo DNA/RNA damage-induced apoptosis caused by failing a cell cycle checkpoint. Additionally, drug-induced perturbations may result in incorrect progression through the cell cycle, which causes the cell to enter a permanent arrested state or die as described above. Note that though RNA damage is known to cause cell cycle arrest and apoptosis (*Bellacosa and Moss, 2003*), the mechanisms are poorly known, so we made the conservative decision to check the level of RNA damage at the same checkpoints as DNA damage.

As an example, we modelled 5-FU-induced RNA and DNA damage in the intestinal epithelium. We considered the two main downstream metabolites of 5-FU, FdUTP and FUTP, causing DNA and RNA damage, respectively (*Longley et al., 2003*). To do this, we implemented in the ABM a previously published model that describes 5-FU distribution post-dosing in mouse and a reduced version of the 5-FU metabolic pathway (*Gall et al., 2023*).

Furthermore, we implemented the effect of FdUTP and FUTP on DNA and RNA synthesis, respectively, on each cell of our ABM using a Hill function as follows:

$$\frac{dDNA_i}{dt} = k_{DNA} - d_{DNA_i} \cdot DNA_i \cdot \frac{FdUTP^n}{FdUTP^n + K_{FdUTP}^n},$$

$$\frac{dRNA}{dt} = k_{RNA} - d_{RNA} \cdot RNA \cdot \frac{FUTP^m}{FUTP^m + K_{FUTP}^m}.$$

Parameter values can be found in *Appendix 1—table 1*.

The impact of these metabolites on DNA and RNA of each cell of the epithelium resulted in the arrest of the majority of proliferative cells, with a small proportion undergoing apoptosis after failing the G1/S or G2/M checkpoint.

## Mechanical cell interactions and contact inhibition

Intestinal stem cells and early progenitor cells compete for limited niche space and, therefore, the ability to retain or regain stemness. Cell proliferation creates a constant battle for space, inducing forces that drive cell migration away from the hard boundary of the stem cell niche towards the top of the crypt and onto the villus.

We assumed intercellular physical forces based on Hertzian contact mechanics with adhesive and frictional forces, similar to those in published reports (*Galle et al., 2005*; *Buske et al., 2011*). For the sake of simplicity and differently from previous approaches, we did not include the extra repulsive force opposing the reduction in cell volume caused by cell overlapping and did not consider radial expansion of cells to compensate for the loss of volume in compressed cells.

In our model, cells experience repulsive, adhesive, and frictional forces. Forces result in movement according to Stoke's flow, where viscous forces dominate inertial forces, such that cell velocity is directly proportional to the resultant forces on the cell. For very shallow overlapping distances ($\lesssim 10\%$ of the cells radius), the adhesive force holds the cells together and replicate continuity of a biological tissue, but for greater overlap distances, repulsive forces dominate. Frictional forces help create collective movement by counteracting cell migration in the opposite direction to the general flow of cells.

All distances are expressed in arbitrary units (A.U.) defined such that 1 distance unit is equal to the diameter of an average, isolated cell. Forces are then measured in the resulting units. We have assumed cells are deformable and hence can lose their spherical shape when responding to mechanical forces. Regions with high proliferation result in cell diameters, in both the z-axis direction (longitudinal crypt–villus axis) and the x–y plane (crypt transversal circumference), smaller than 1 unit and, hence, in inequality between the number of cells and the distance units.

### Contact repulsion

Cells are assumed to be elastic spheres with intercellular forces derived from Hertzian contact mechanics. The magnitude of the repellent force, $F_{rep.}^{ij}$ between cell $i$ (with position vector $\mathbf{x}_i$, radius $R_i$, Young's modulus $E_i$, and Poisson ratio $\nu_i$) and cell $j$ (with position vector $\mathbf{x}_j$, radius $R_j$, Young's modulus $E_j$, and Poisson ratio $\nu_j$) is described as follows:

$$F_{rep.}^{ij} = -\frac{4}{3} E^* R^{\frac{1}{2}} d_{ij}^{\frac{3}{2}}, \quad \frac{1}{E} = \frac{1-\nu_i^2}{E_i} + \frac{1-\nu_j^2}{E_j}, \quad \frac{1}{R} = \frac{1}{R_i} + \frac{1}{R_j},$$

where $d_{ij} = R_i + R_j - |\mathbf{x}_{ij}|$ is the overlapping distance between cells measured on the line joining the cell centres, with $\mathbf{x}_{ij} = \mathbf{x}_j - \mathbf{x}_i$ the displacement vector joining the two cell centres. This repulsive force acts on both cells in opposing directions, pushing them away along the unit vector joining the two cells $\hat{\mathbf{x}}_{ij}$:

$$\mathbf{F}_{rep.}^{ij} = F_{rep.}^{ij} \hat{\mathbf{x}}_{ij}.$$

The reported value for the Young's modulus of Paneth cell is relatively large (*Pin et al., 2015*) and results in a relatively large force acting on neighbouring stem cells which helps to confine them

in the niche. In addition, the previously published values of the Poisson ratio indicate that cells are marginally compressible (*Geissler and Hecht, 1981*).

## Adhesive force

All cells in contact experience adhesive forces proportional to the area of contact and the cells inherent adhesiveness, parameterized by $\epsilon$. The magnitude of adhesive force between cell $i$ and $j$ is quantified as follows:

$$F_{adh.}^{ij} = 2\epsilon\pi p_{ij}\left(1 - \frac{p_{ij}}{|\mathbf{x}_{ij}|}\right),$$

where $|\mathbf{x}_{ij}|$ is the distance between cell centres and

$$p_{ij} = \frac{R_i^2 - R_j^2 + |\mathbf{x}_{ij}|^2}{2|\mathbf{x}_{ij}|}.$$

This force is again directed along $\hat{\mathbf{x}}_{ij}$, pulling the cells together: $\mathbf{F}_{adh.}^{ij} = F_{adh.}^{ij}\hat{\mathbf{x}}_{ij}$ and its magnitude is derived by assuming the associated energy, $E_{adh.}^{ij}$, is proportional to the area of contact between cells $i$ and $j$, $E_{adh.}^{ij} = \epsilon A^{ij}$, where $A^{ij} = \pi\left(R_i^2 - p_{ij}^2\right)$, and differentiating with respect to the distance between the cells.

Two cells in isolation will be at rest when the repulsive and adhesive forces are equal; however, in our simulations, this rarely happens due to the constant proliferation and growth of surrounding cells. In vivo crypts have a highly compressed niche with tightly packed stem cells wedged between Paneth cells. In our model, the repulsive force is parameterized entirely by observed quantities (the Young's modulus and Poisson ratio), leaving $\epsilon$ in the adhesive force as a free parameter. The value of $\epsilon$ determines intercell separation at rest. This value was chosen to allow overlapping of Paneth cells at rest of 0.15 distance units, which corresponds to 15% of the diameter of an average Paneth cell. This results in $\epsilon = 0.216$ for Paneth–Paneth adhesion. Qualitatively, all other cells are less tightly packed, so all other adhesive forces (including Paneth cells with any other cell type) are assumed to be tenfold weaker with $\epsilon = 0.0216$, which produces an overlap of approximately 0.075 cell units. These assumptions facilitate the recapitulation of the tighter packed cells in the niche, resulting in increased mechanical pressure (defined in the following sections) which induces proliferation contact inhibition mechanisms.

## Frictional force

Cells that are in contact experience a frictional force proportional to their relative velocity. The force acting upon cell $i$ due to friction with cell $j$ is quantified as follows:

$$\mathbf{F}_{fric.}^{ij} = \mu_{fric.}A^{ij}\left(\frac{d\mathbf{x}^i}{dt} - \frac{d\mathbf{x}^j}{dt}\right),$$

where $A^{ij}$ is the area of contact between cells $i$ and $j$ defined above, and $\mu_{fric.}$ is a numerical constant calibrated to enforce orderly cell dynamics. This force is comparatively smaller than the other forces but helps the collective motion of cells by opposing cell migration against the common direction.

## Cell migration

Under a force, cells move according to Stoke's flow, where viscous forces are assumed to dominate over inertial effects:

$$m\frac{d^2\mathbf{x}^i}{dt^2} = \sum_j \mathbf{F}^{ij} - \mu\frac{d\mathbf{x}^i}{dt} \Rightarrow \frac{d\mathbf{x}^i}{dt} = \frac{1}{\mu}\sum_j \mathbf{F}^{ij}.$$

Therefore, the position vector of the $i$-th cell, $\mathbf{x}^i$, is updated according to

$$\Delta\mathbf{x}^i = \frac{1}{\mu}\sum_j \mathbf{F}^{ij}\Delta t,$$

where $\mathbf{F}^{ij} = \mathbf{F}^{ij}_{rep.} + \mathbf{F}^{ij}_{adh.} + \mathbf{F}^{ij}_{fric.}$ is the resultant of all forces on cell $i$ due to cell $j$.

The parameter $\mu$ links the forces to cellular motion. The value of this parameter is estimated to recapitulate the transfer velocity in the crypt–villus junction measured in in vivo experiments to be approximately one cell position per hour in mice (**Potten, 1998**). However, cell motion response to these forces may vary for different cell types. It has been reported that Paneth cells persist in the stem cell niche at the crypt base for relatively long periods of up to 57 d in mice (**Ireland et al., 2005**; **Roth et al., 2012**) and exhibit elevated $\beta_4$-integrin expression anchoring them to the mesenchyme (**Langlands et al., 2016**). Additionally, Paneth cells are larger and stiffer than the comparatively malleable stem cells which suggest that they require greater forces to be displaced. In our model, we used μ to replicate this behaviour and recreate drag effects of the basal membrane/mesenchyme. We implemented a value of $\mu$ for Paneth cells 10,000-fold greater than for other cells, effectively making Paneth cells difficult to move by other cells but allowing them to slowly move one another to form an orderly niche over longer timescales.

## Internal pressure and contact inhibition

The forces described above are used to calculate the internal pressure experienced by cells, which varies according to the cell-intrinsic properties and local environment, that is, a stem cell in the crowded niche has higher internal pressure. Cell pressure is used to recapitulate contact inhibition by modulating the duration of the division cycle which increases when cells are densely squeezed together and decreases if cell density falls to enable, for instance, fast recovery from injury.

A cell feels internal stress from the surrounding cells, and this is used to simulate contact inhibition. To do this we use the concept of virial stress outlined in **Van Liedekerke et al., 2015**. The stress tensor for cell $i$, $\sigma_i$, is defined as follows:

$$\sigma_i = \frac{1}{V_{cell}} \sum_j \mathbf{F}^{ij} \otimes \mathbf{r}^{ij},$$

where $\mathbf{r}^{ij}$ is the vector from the centre of the cell $i$ to the plane of contact with cell $j$, always assumed to be on the surface of cell $i$, and $\otimes$ is the tensor/outer product combining two vectors into a 'matrix'. Using this stress tensor, we extract the pressure in the conventional manner:

$$p = -\frac{1}{3} tr\left(\sigma\right).$$

As all our forces are normal to the plane of contact, this reduces to

$$p = \frac{1}{4\pi R_i^2} \sum_j |\mathbf{F}^{ij}|.$$

This provides a rough, first-order approximation to the pressure experienced at the centre of the cell that is straightforward to compute and essential to implement contact inhibition in proliferative cells. Note that we do not consider the hydrostatic pressure induced by cell compression.

On the other hand, physical compression has been reported to lead to YAP inactivation, retarding growth and morphogenesis in the GI epithelium (**Halder et al., 2012**; **Aragona et al., 2013**; **Low et al., 2014**). We used our estimate of pressure to implement this contact proliferation inhibition mechanism responding to environmental mechanical cues and described the increase in the cell cycle duration, $t_{cycle}$, as pressure, $p$, increases using a scaled logistic function as follows:

$$t_{cycle} = \frac{g}{\left(1 + e^{2\left(p_0 - p\right)}\right)} + t^{short}_{cycle}.$$

Here, $p_0$ is the average pressure experienced by cells in the niche; $t^{short}_{cycle}$ is the average division time of absorptive progenitors; and $g = 2\left(t^{long}_{cycle} - t^{short}_{cycle}\right)$, where $t^{long}_{cycle}$ denotes the longer division time of a stem cell in average niche conditions.

This function captures the variation of the duration of the division cycle from a minimum to a maximum value in highly compressed cells which leads to longer division times in the tightly constrained stem cell niche of the crypt, while the cycle is shorter in the less compressed transit-

amplifying zone, in agreement with experimental reports (*Wright and Alison, 1984*; *Bach et al., 2000*; *Schepers et al., 2011*).

## Biochemical signalling

Next, we detail how the cells interact with one another, communicating the local composition of the crypt to maintain homeostasis through simulated biochemical signalling.

To achieve stable crypt cell composition and structure, we have implemented five signalling mechanisms including Wnt, Notch, and BMP pathways which have been demonstrated to be essential for morphogenesis and homeostasis of the intestinal crypt (*Gehart and Clevers, 2019*; *Fevr et al., 2007*; *VanDussen et al., 2012*; *Pellegrinet et al., 2011*; *He et al., 2004*). We have modelled contact proliferation inhibition mediated by the YAP-Hippo signalling pathway responding to mechanical forces (*Halder et al., 2012*; *Aragona et al., 2013*; *Low et al., 2014*) as described above and following experimental evidence (*Hao et al., 2012*; *Koo et al., 2012*; *Farin et al., 2016*), implemented a ZNRF3/RNF43-like mediated feedback mechanism between Paneth and stem cells.

These minimal signalling mechanisms were chosen because a full understanding of the protein interaction networks is still a topic of active research. However, even with our conservative assumptions, we implicitly introduce crosstalk between the different signalling pathways. For example, the nature of cell fate decisions leads to interaction between Wnt and Notch levels, and changes in the duration of the cell cycle caused by contact inhibition regulate the ability of a cell to accumulate signalling molecules.

### Wnt signalling

The Wnt pathway is the primary pathway associated with stem cell maintenance and differentiation in the intestinal crypt as well as in many other tissues (*Fevr et al., 2007*; *van der Flier and Clevers, 2009*; *Nusse and Clevers, 2017*). Two sources of Wnt signals have been described in the mouse crypt: Paneth cells (*Sato et al., 2011*) and specific mesenchymal cells surrounding the stem cell niche at the crypt base (*Stzepourginski et al., 2017*).

We did not consider the dynamics of the canonical Wnt signalling molecular cascade but directly implemented downstream cellular responses to Wnt levels. We modelled Wnt signalling as a short-range field around Paneth cells and Wnt-emitting mesenchymal cells at the bottom of the crypt, acting within a distance *WntRange* from the surface of these cells (see *Appendix 1—table 1* for value). Receptive cells within this range tether Wnt signals to their surface as previously reported (*Farin et al., 2016*; *Clevers and Nusse, 2012*). This is described by the following equation:

$$\frac{dWnt}{dt} = \begin{cases} k_{Wnt} Inc_{Wnt} - d_{Wnt} \left( ZNRF3 \right) Wnt & Wnt < M_{Wnt}, \\ -d_{Wnt} \left( ZNRF3 \right) Wnt & Wnt \geq M_{Wnt}. \end{cases}$$

The variable '*Wnt*' is an abstraction of the total number of Wnt ligands tethered to the surface of the cell. $k_{Wnt}$ is the rate of Wnt signal tethering by a receptive cell and $d_{Wnt} \left( ZNRF3 \right)$ is the decay rate of Wnt signal tethered molecules. $d_{Wnt}$ depends on the turnover of Wnt receptors assumed to be regulated by RNF43 and ZNRF3 ligands produced by stem cells, which forms a Wnt-mediated negative feedback loop as described below. $M_{Wnt}$ describes the maximum number of Wnt signals a cell can have tethered and its value is chosen to be a power of 2 to facilitate dividing Wnt signals in half upon cellular division. $Inc_{Wnt}$ is the total amount of Wnt signal sources within range of the cell and is calculated as follows:

$$Inc_{Wnt} = \sum \text{Paneth in range} + \frac{Mesenchymal_{niche}}{Cells_{niche}}.$$

$Mesenchymal_{niche}$ represents the number of Wnt-emitting mesenchymal cells surrounding the niche, which we assume is equal to the total number of epithelial cells in the niche in homeostatic conditions (*Wright and Alison, 1984*; *Snippert et al., 2010*). Additional Wnt production by Paneth cells is required to support the homeostatic number of stem cells in homeostasis. In the presented modelling scenarios, we assumed constant exogenous Wnt source, that is, constant $Mesenchymal_{niche}$, shared by all cells in the niche and enhancing niche recovery after damage. For instance, with lower number of cells in the niche, the survival cells will receive stronger mesenchymal Wnt signalling that

enhances proliferation and recovery after perturbations. We assumed that surface-tethered signals are equally distributed between daughter cells upon cell division (*Gehart and Clevers, 2019*; *Farin et al., 2016*), so that cells eventually lose Wnt signals and their capacity to proliferate if not within the range of a Wnt source. These assumptions are partly supported by observed in vivo and in vitro behaviour, where the mesenchymal and Paneth cell-derived Wnt sources are mutually redundant (*Farin et al., 2012*).

### ZNRF3/RNF43 signalling

In our model, Paneth cells enhance their own production by generating high Wnt local environments (*van Es et al., 2005*). In addition, due to their high Young's modulus, Paneth cells create a region of high intercellular forces on neighbouring cells which leads to prolonged division times with greater opportunity for Wnt accumulation. This, in turn, expands the niche region with high Wnt and high cell pressure, promoting further differentiation into stem and Paneth cells. Therefore, without a negative feedback mechanism in our model, these features would result the expansion of the niche with stem and Paneth cells occupying the entire crypt. Additionally, two recent studies have demonstrated the existence of a negative feedback loop mediated by RNF43 and ZNRF3 ligands produced by stem cells (*Hao et al., 2012*; *Koo et al., 2012*). These studies proposed that RNF43 and ZNRF3 inhibit Wnt signalling by promoting the turnover of Wnt receptors such as Frizzled and LRP5 (*de Lau et al., 2011*), and showed that simultaneous deletion of these two receptors results in the formation of adenomas comprising mostly stem and Paneth cells (*Koo et al., 2012*).

We assumed that ZNRF3/RNF43 (henceforth called ZNRF3 for simplicity) is a diffusing, decaying signal secreted by stem cells. Without explicit knowledge of the chemical and physical properties of ZNRF3 signalling, this process is assumed to immediately reach steady state at the timescale of cellular decisions. Therefore, the ZNRF3 signal strength, $ZNRF3\left(\mathbf{r}\right)$, received by a cell at position $\mathbf{r}$ from a stem cell located at position $\mathbf{R}$, is described by the diffusion equation as follows *Crank, 1975*:

$$ZNRF3\left(\mathbf{r}\right) = Ze^{\dfrac{-\left|\mathbf{R}-\mathbf{r}\right|}{L_{ZNRF3}}} \ ,$$

where $Z$ represents the maximum signal strength immediately around the emitting cell and $L_{ZNRF3}$ determines the spatial scale of diffusion, which we assume is equal to the length of a cell in order to maintain high signalling levels primarily in the niche.

The total ZNRF3 signalling received by a cell at position $\mathbf{r}$ is calculated, therefore, as the sum of the signal received from all stem cells:

$$ZNRF3\left(\mathbf{r}\right)_{total} = \sum_{\text{stem cells}} Ze^{\dfrac{-\left|\mathbf{R}_i-\mathbf{r}\right|}{L_{ZNRF3}}} \ ,$$

where $\mathbf{R}_i$ is the position of the $i$-th stem cell.

The strength of ZNRF3 signalling received by a cell is proportional to the number of stem cells in the immediate vicinity of the cell: in typical, homeostatic conditions, $ZNRF3\left(\mathbf{r}\right)_{total}$ is high in the niche, falling off exponentially as a cell moves towards the villus.

The ZNRF3 signalling level detected by a cell, located at position $\mathbf{r}$, regulates the decay rate of its surface-tethered Wnt molecules, $d_{Wnt}$ as follows:

$$d_{Wnt}\left(ZNRF3\right) = \dfrac{K}{1+\left(\dfrac{ZNRF3}{ZNRF3\left(\mathbf{r}\right)_{total}}\right)^{\mathrm{u}}},$$

where $u$ is a scaling coefficient, and $K$ and $ZNRF3_*$ are constants calibrated to maintain the size of niche at its homeostatic level. In particular, $ZNRF3_*$ is determined by the homeostatic number of stem cells in the niche (*Snippert et al., 2010*), while $K$ was calibrated to produce a Wnt decay rate high enough to prevent Wnt values ≥64 in cells located at the edge of the niche when the number of stem cells is excessive such that $ZNRF3(\mathbf{r})_{total} \geq ZNRF3_*$. These considerations prevent the expansion of the niche by preventing cells from differentiating into the Paneth or stem cell fate (which requires $Wnt \geq 64$) when a cell is outside the niche.

With this implementation of ZNRF3-mediated negative feedback, the Wnt decay rate within the niche is high but is compensated by the abundant Wnt supply from mesenchymal and Paneth sources, while the Wnt decay rate rapidly drops to zero outside the niche. This means that the degradation of Wnt outside the niche has little impact on a healthy crypt and the Wnt gradient in our model is mainly generated by the halving of the surface bound Wnt signals between daughter cells upon division. Growth and proliferation derived forces drive migration of cells towards the villus while the amount of tethered Wnt decreases after each division. This recreates the observed (*Farin et al., 2016*) decreasing gradient of Wnt signals moving up the crypt (*Figure 1*), with the highest values in the niche, intermediary values in the transit-amplifying zone, and low levels in the upper crypt region of differentiated enterocytes.

The stem cell-mediated negative feedback loop regulating Wnt signalling, together with the differentiation rules described below, ensures the maintenance of the niche size and crypt composition in homeostasis. In addition, it also facilitates crypt recovery as stem cells in low numbers are able to reach greater surface-tethered Wnt levels to pass to their offspring which, in turn, can more readily acquire the required amount of Wnt to become stem cells.

## Notch signalling

Active Notch signalling requires direct membrane contact between two cells, one expressing Notch ligands and the other Notch receptors (*Gehart and Clevers, 2019*; *Pellegrinet et al., 2011*; *Baron, 2003*; *Sancho et al., 2015*). In the intestinal epithelium, Notch ligands present in secretory cells bond to transmembrane notch receptors of stem cells to induce a transcriptional cascade which blocks differentiation of stem cells into the secretory lineage in a process known as lateral inhibition and leads to checkerboard/on-off pattern of Paneth and stem cells in the niche (*VanDussen et al., 2012*; *Chen et al., 2017*). With these considerations, Notch signalling, $Notch$, is implemented in each cell according to the following equation:

$$\frac{dNotch}{dt} = k_{Notch} \left( Inc_{Notch} - Notch \right),$$

where $Inc_{Notch}$ is the amount of incoming notch ligands to the cell which we assumed is equal to the number of ligands expressing cells in contact with the cell. At steady state, a cell's Notch value corresponds to the number of incoming Notch ligands the cell is receiving: for example, a stem cell receiving Notch from one single neighbouring cell reaches equilibrium with $Notch = 1$. The factor $k_{Notch}$ denotes the rate of Notch accumulation and has a relatively high value to ensure that the equilibrium is reached before the fate-commitment point at the end of G1. As described in the cell cycle section, the duration of G1 changes with the length of the overall division cycle: shorter cycles have a shorter G1 phase, shortening the time the cell has to receive Notch signals before deciding whether to differentiate or divide. Additionally, $k_{Notch}$ is also the decay rate of the cell Notch signalling and this relatively fast rate means that Notch must be constantly supplied for a stem cell to maintain stemness.

A reduction in cell density (e.g. by ablation of cells) can introduce gaps in the simulated epithelial tissue. In real tissues, these gaps would be covered by expansion-flattening of surviving cells to restore epithelial integrity and contact to neighbouring cells. These new contacts would allow cells to exchange Notch ligands. In our model, we do not explicitly consider the expansion of cells to fill gaps in the epithelium; however, we simulate this effect by allowing a cell to pass Notch signals to receiving cells within a larger range (one cell diameter) following a drop in local density. This allows our model to recreate the correct recovery response following ablation of cells.

## BMP signalling

The Wnt gradient in the crypt is opposed by a gradient of BMP generated by mesenchymal telocytes, which are especially abundant at the villus base and provide a BMP reservoir, and by the recently identified trophocytes located just below crypts and secreting the BMP antagonist Gremlin1 (*McCarthy et al., 2020*). BMP signals inhibit cell proliferation and promote terminal differentiation (*Qi et al., 2017*). Large levels of BMP at the crypt–villus junction prevent proliferative cells from reaching the villus (*Beumer et al., 2022*). BMP signalling has been reported to be modulated by matured epithelial cells on the villus via hedgehog signalling (*Büller et al., 2012*; *van den Brink et al., 2004*) such that a decrease in villus cells decreases BMP signalling in the crypt, which enhances proliferation and expedites villus regeneration.

We propose a simple model that assumes that enterocytes, $E$, secrete diffusing signals, which could be interpreted as Indian hedgehog, to regulate BMP secretion by mesenchymal cells. The explicit pathways and associated timescales involved in BMP signalling are unknown; therefore, similar to our implementation of ZNRF3 signalling, this process is assumed to instantaneously reach steady state at the timescale of cellular decisions. As before, we assume that BMP is a diffusing, decaying signal in steady state (*Crank, 1975*) described by

$$BMP\left(z, E\right) = f\left(E\right) e^{-ln(B) \frac{\left(z_{top}\left(E\right) - z\right)}{\left(z_{top}\left(E\right) - z_{50}\right)}},$$

where $z$ is the position coordinate corresponding to the crypt–villus longitudinal axis; in our model $z \leq 0$ for cells located in the stem cell niche while $z > 0$ for crypt cells outside the niche; $z_{top}\left(E\right)$ is the value of $z$ at the top of the crypt, which depends on the number of enterocytes on the villus; $B$ is the exponential transformation of the diffusion coefficient. To facilitate the use of the model for different species, the $z$ coordinate is standardized using $z_{50}$, which is the crypt axis position at which the number of mature enterocytes becomes greater than the number of absorptive progenitors. As mentioned above, mesenchymal cells surrounding the niche secrete BMP antagonists (*McCarthy et al., 2020*), and we assumed that BMP signalling is effectively blocked in the niche such that $BMP = 0$, which is approximately true for the above formula. $f\left(E\right)$ describes the relationship between the number of enterocytes and maximum BMP signal intensity using an increasing Hill function:

$$f\left(E\right) = \frac{2A}{1 + \left(\frac{E_h}{E}\right)^p},$$

where $E_h$ is the homeostatic number of enterocytes determined by in vivo experiments, $p$ is the Hill interaction coefficient, and $A$ denotes the level of BMP signals at position $z_{top}$. In our model, absorptive progenitors differentiate into enterocytes when $BMP > Wnt$, representing that the anti-proliferative BMP signalling received by the cell is sufficient to overcome the proliferative effect of Wnt (*He et al., 2004*). We achieved a homeostatic crypt cell composition with values of $A$ and $B$ that allow progenitors cells to divide in a healthy crypt at least three times before differentiating. Differentiation occurs when the Wnt content of a cell, at position $z$, reaches values below $BMP\left(z\right)$ when migrating towards the villus.

In addition, these equations describe a frequently reported feedback response to villus injury consisting of enhanced proliferation within hypertrophic crypts (*Büller et al., 2012*; *Pont and Yan, 2018*; *Sprangers et al., 2021*). In our model, when the number of enterocytes on the villus falls below the homeostatic level, the production of BMP signals decreases and makes it possible for absorptive progenitors to divide more times and reach higher positions in the crypt before becoming terminally differentiated. Concurrently, the height of the crypt must increase to provide sufficient space for the extra proliferative cells. We modelled the enlargement of the crypt height responding to villus injury by varying the maximum $z$-coordinate of the crypt, $z_{top}$, using a decreasing Hill function as follows:

$$z_{top}\left(E\right) = \frac{C_h z_0}{1 + \left(C_h - 1\right)\left(\frac{E}{E_h}\right)^q},$$

where $z_0$ is the calibrated homeostatic value of $z_{top}$, $C_h$ is the maximum fold increase in the height of the crypt, and $q$ the Hill interaction coefficient. We do not consider cases in which the number of enterocytes on the villus increases above homeostatic levels, such that if $E > E_h$ then $z_{top}\left(E\right) = z_0$.

The standard manner to report the height of cells along the crypt–villus axis is in terms of cell positions, which is related to but not equal to $z_{top}$. This is because cell positions are counted from the bottom of the niche (and we defined $z = 0$ to be the top of the niche), and that in our model the cells are squeezed together, causing the height of the crypt measured in cell positions to be larger than $z_{top}$.

## Cell fate: Proliferation, differentiation, arrest, and apoptosis

In the sections above, we have outlined the dynamics of signalling pathways, cell cycle proteins, and mechanical forces. These processes interact with each other to maintain epithelial homeostasis by precisely tuning cell proliferation, differentiation, and migration within the crypt geometry.

An overall picture integrating the rules governing cell fate decision is described in *Figure 1*. Wnt levels ≥64 A.U. are required for stemness maintenance. For a stem cell, lateral inhibition is repressed when Notch < 3 A.U., equivalent to less than three secretory cells in the local neighbourhood. If Notch is repressed (<3 A.U.) and Wnt > 64 A.U., stem cells differentiate into Paneth cells. Paneth cells generate Wnt signals which enhance the production of stem cells and of Paneth cells themselves. Niche expansion is modulated by the ZNRF3/RNF43-mediated negative feedback mechanism (*Hao et al., 2012*; *Koo et al., 2012*; *Farin et al., 2016*) that makes Wnt > 64 unobtainable after reaching the homeostatic number of stem cells. Furthermore, the duration of the division cycle is dependent on local forces experienced by the cell. Cells under high mechanical pressure (in the niche) are subjected to YAP-Hippo-regulated contact inhibition and with longer cycles accumulate more Wnt and Notch signals. On the other hand, cells located outside the niche exhibit shorter cycles and cannot effectively accumulate enough Wnt signals to become stem or Paneth cells.

Stem cells with decreased levels of Wnt signalling (<64), usually located outside the niche, differentiate into absorptive proliferating progenitors if Notch signalling is active or into secretory progenitors if Notch signals <2 A.U. This lower Notch threshold value is required to maintain the correct balance of absorptive and secretory cells outside the niche in the absence of large numbers of Notch secreting Paneth cells. All cells migrate towards the crypt mouth driven by proliferation forces. During this migration, the Wnt content in absorptive progenitors is halved in each division and, away from Wnt sources, progressively decreases, while BMP signals increase, towards the villus. In our model, differentiation into enterocytes occurs when progenitors encounter a BMP signal level higher than their Wnt signal content. For instance, in the ileal crypt in homeostasis, this occurs approximately at cell position 16 from the crypt base, where progenitors migrating from the stem cell niche reach a reduced content of Wnt signals of about 8 A.U. On the other hand, the BMP signalling level has a maximum value of 64 at approximately cell position 23 from the crypt base, where BMP signals are generated by mature enterocytes. These BMP signals diffuse towards the crypt base and, hence, decrease exponentially to reach values of 8 A.U. at approximately position 16, which enables differentiation into enterocytes. Epithelial injuries resulting in a decreased number of enterocytes reduce BMP signal production and its diffusion range which results in the enlargement of the proliferation compartment as cells encounter the required level of BMP signals for differentiation only at higher positions in the crypt.

All fate decisions are assumed to be made at the restriction point which in our model is located at the end of G1 (*Blomen and Boonstra, 2007*). At the restriction point, cells assess their internal Wnt and Notch levels, and if these values fulfil the criteria to differentiate, they enter a quiescent state or G0, otherwise they proceed to S-phase and become irreversibly committed to complete the cell cycle of variable duration depending on local forces. This quiescent state lasts for 4 hr for all differentiating cells, except for absorptive progenitors, which differentiate straightaway into enterocytes. In accordance with *Stamataki et al., 2011*, a secretory progenitor requires an additional 4 hr to fully mature into a goblet or enteroendocrine cell.

Therefore, quiescent stem cells located above the fourth cell position from the crypt base (*Gehart and Clevers, 2019*; *Potten et al., 1978*; *Sangiorgi and Capecchi, 2008*) emerge naturally in the model as stem cells migrate outside the niche and pause the cycle to give rise to non-proliferative secretory progenitors, which have been identified with quiescent stem cells (*Buczacki et al., 2013*; *Clevers, 2013a*). Features and behaviours of these cells could be expanded if of interest for the model application.

Cell fate decisions are reversible; a stem cell that leaves the niche and differentiates into a progenitor cell can relatively quickly become a stem cell again if regaining enough Wnt signals by being pushed back into the niche. This plasticity extends to all cells: all progenitors and fully differentiated cells can revert to stem cells when exposed to sufficient levels of Wnt and Notch signals, replicating injury recovery mechanisms observed in the crypt (*Hageman et al., 2020*; *Tetteh et al., 2016*). We have assumed that all cells, except Paneth cells, need to acquire and maintain high levels of Wnt signals (>64) over 4 hr to complete the process. Dedifferentiating cells shrink to their new smaller size during the process if required.

Notch signalling mediates the process of Paneth cell de-differentiation into stem cells to regenerate the niche as previously reported (*Mei et al., 2020*; *Yu et al., 2018*). Paneth cells not

supplying Notch ligands for 12 hr to recipient cells dedifferentiate into stem cells in a process that takes 36 hr to complete in agreement with published findings (*Yu et al., 2018*).

Additionally, Paneth cells in low Wnt conditions (e.g. a Paneth cell that is forced out of the niche) for 48 hr will also dedifferentiate into a stem cell, which with low Wnt content rapidly becomes a secretory or absorptive progenitor.

Additionally, injured proliferative cells can experience cell cycle arrest and apoptosis, induced by drug injury or by natural senescence. In arrested and apoptotic proliferative cells, the production rates of the cyclins ($V_{sa}$, $V_{sb}$, $CycD_0$, and $V_{se}$) are set to 0 to interrupt the cell cycle. We assumed that cells remain arrested until they are shed from the villus tip or reach the end of their lifespan and become apoptotic. Apoptotic cells shrink and die with a negative linear rate of

$$\frac{dr}{dt} = -\frac{r_{apop.}}{t_{apop.}},$$

where $r_{apop.}$ is the radius at the onset of apoptosis and $t_{apop.}$ is the time for the completion of apoptosis.

## ABM simulation of Ki-67 and BrdU staining

This section discusses the implementation of the Ki-67 and BrdU staining simulations, which can be found in *Figures 3 and 5*, and is discussed in the 'Results' section.

For the Ki-67 staining simulation, we considered that a cell is Ki-67 positive if it is going through S-, G2-, or M-phase of the division cycle. Daughter cells are considered Ki-67 positive, regardless of their fate, during the first 6 hr after cell division. This assumption recapitulates the time reported for the Ki-67 protein to decay below detectable levels after exiting the cycle (*Miller et al., 2018*) and the detection of Ki-67 in G1 in continuously cycling cells (*Sobecki et al., 2017*). Similarly, cells are assumed to remain Ki-67 positive for 6–12 hr after drug-induced cell cycle interruption depending on the phase the cell was in upon interruption in our simulations, which recapitulates a previously published report (*Miller et al., 2018*), where cells exhibited greater Ki-67 levels in later cell cycle phases. In particular, cells arrested during G1, S, G2, and M phases are Ki-67 positive for 6, 8, 10, and 12 hr after arrest, respectively.

For the BrdU staining simulation, we assumed that cells become BrdU positive by effectively incorporating BrdU into their DNA when they are in S-phase or enter S-phase during the BrdU exposure window, which is considered to last 2 hr after BrdU administration in agreement with previous experimental reports (*Parker et al., 2017*). The initial level of BrdU after dosing in each cell is quantified by

$$BrdU = Round\left(K_{BrdU}\frac{T_{BrdU}}{2}\right),$$

where $Round(X)$ is the function that rounds $X$ to the nearest integer, $K_{BrdU}$ is the theoretical maximum level of BrdU a cell can incorporate, and $T_{BrdU}$ is the remaining BrdU exposure time. For cells that enter S-phase after BrdU administration, $T_{BrdU}$ is equal to the remaining BrdU exposure time, while for cells already in S-phase at the time of BrdU dosing, $T_{BrdU}$ is equal to the exposure window or, alternatively, to the remaining duration of S-phase if this is shorter than the exposure window. Furthermore, we considered that a cell is BrdU positive if its BrdU level is >0 and, if dividing, the two daughter cells are given a BrdU value of $BrdU_{daughter} = BrdU_{parent} - 1$. This consideration recapitulates experimental reports indicating that the BrdU cell content is diluted in each division and it is no longer detected after 4–5 generations (*Wilson et al., 2008*).

The spatial data from the Ki-67 and BrdU staining experiments comprises the proportion of positive cells at each cell position by aggregating spatial counts from 20 to 50 one-dimensional longitudinal strips running from the crypt base to the villus (*Parker et al., 2017*; *Williams et al., 2016*). Therefore, cell position is reported in a one-dimensional space and measured as the cell count from the base of the crypt to the cell itself. To match these observations, we have implemented an algorithm that slices longitudinally the simulated crypts to generate 100 one-dimensional strips which are aggregated to estimate the proportion of stained cells at each position.

Furthermore, we estimated a 95% confidence interval, based on experimental error, around the simulated spatial profiles of Ki-67- and BrdU-positive cells by assuming that the proportion of

stained cells follows a beta distribution with parameters $\alpha$ and $\beta$, $P \sim Beta\,(\alpha, \beta)$. These parameters are estimated as follows:

$$\hat{\alpha} = p^2 \left( \frac{1-p}{e^2} - \frac{1}{p} \right), \qquad \hat{\beta} = p(1-p) \left( \frac{1-p}{e^2} - \frac{1}{p} \right),$$

where $p$ is the simulated proportion and $e$ its standard error. We used an estimate of the standard error, $e$, derived from experimental data. We first studied the relationship between the mean value and the standard deviation of the proportion in three replicated control samples. The experimental data suggest that the error is lower for extreme values of $p$, that is, around 0 or 1, and larger for values of $p$ around 0.5 (*Appendix 1—figure 3*). Thus, we described this relationship with a quadratic expression:

$$e\,(p) = a_0 + a_1 p + a_2 p^2,$$

where $a_0$, $a_1$ and $a_2$ are the coefficients determined from the replicated control samples and their values are displayed in *Appendix 1—figure 3*.

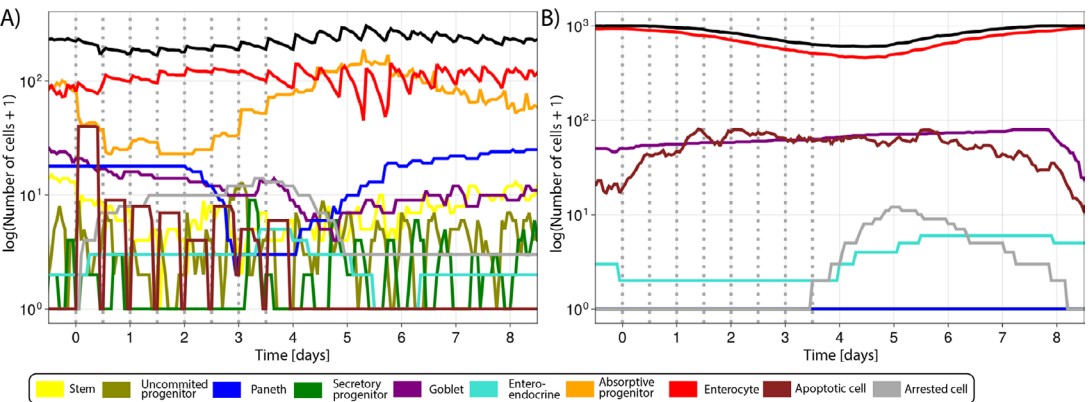

**Appendix 1—figure 1.** Simulated cell lineages in the crypt (**A**) and villus (**B**) during a 4-day CDK1 inhibition treatment for 6 hr, every 12 hr for four consecutive days and following recovery in the agent-based model (ABM) as described in *Figure 4*.

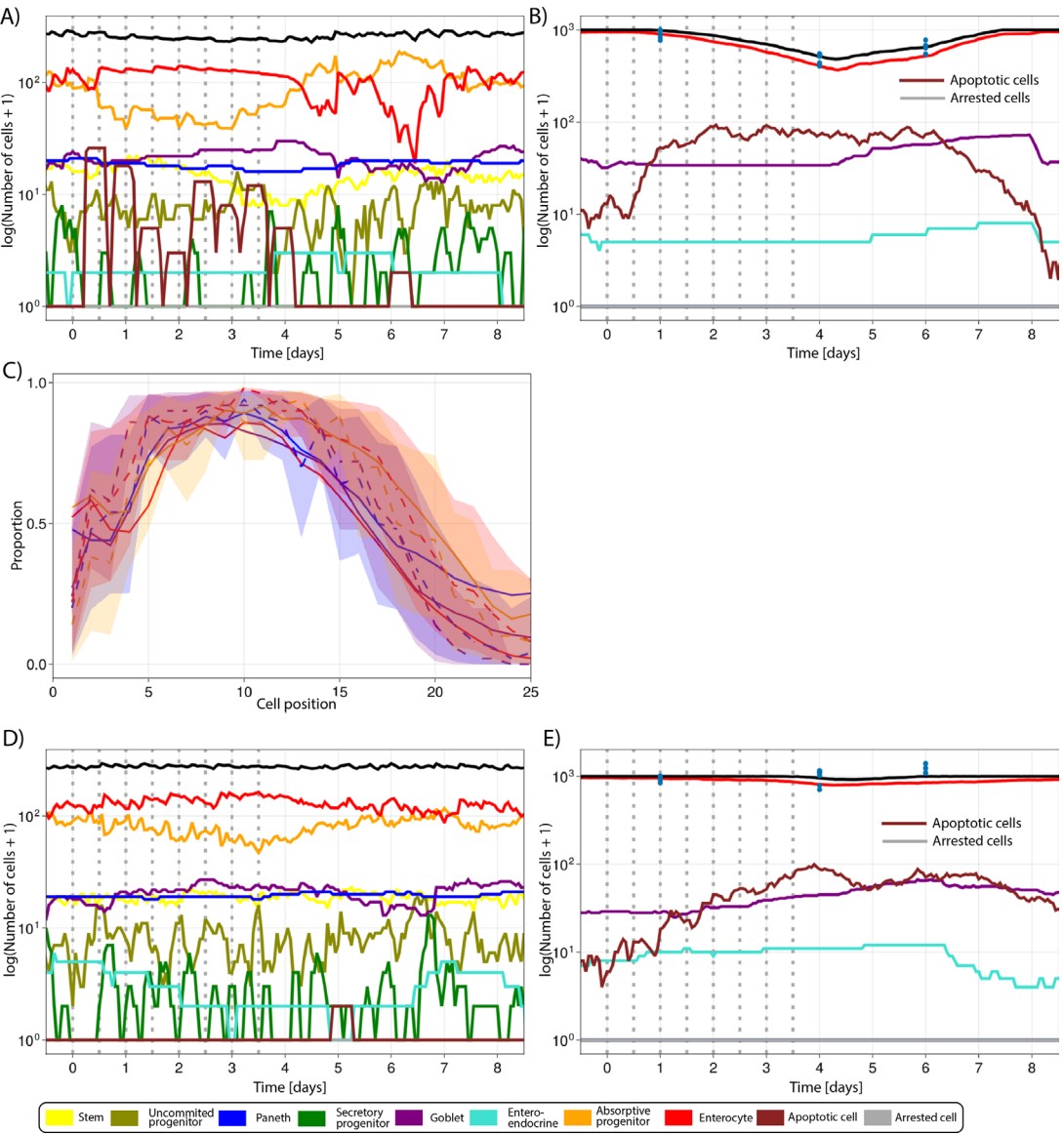

**Appendix 1—figure 2.** Simulated (lines) and observed (symbols) number of cells in the crypt. (**A**) and villus (**B**) during the administration of 50 mg/kg of 5-fluorouracil (5-FU) twice a day for 4 d and following recovery as described in *Figure 5*. (**C**) Predicted (continuous lines) and observed (dashed line) proportions of Ki-67-positive cells along the crypt axis at 6 hr, 1 d, 4 d, and 6 d following the administration of 20 mg/kg of 5-FU twice a day for 4 d. Shadows depict the 95% confidence interval of our simulated staining results assuming that the proportion of staining cells has a beta distribution and estimating its error from experimental data. (**D, E**) Predicted (lines) and observed (symbols) number of cells in the crypt (**D**) and villus (**E**) during the administration of 20 mg/kg of 5-FU twice a day for 4 d and subsequent recovery. Vertical bars represent dosing times. Symbols represent cell counts from individual mice.

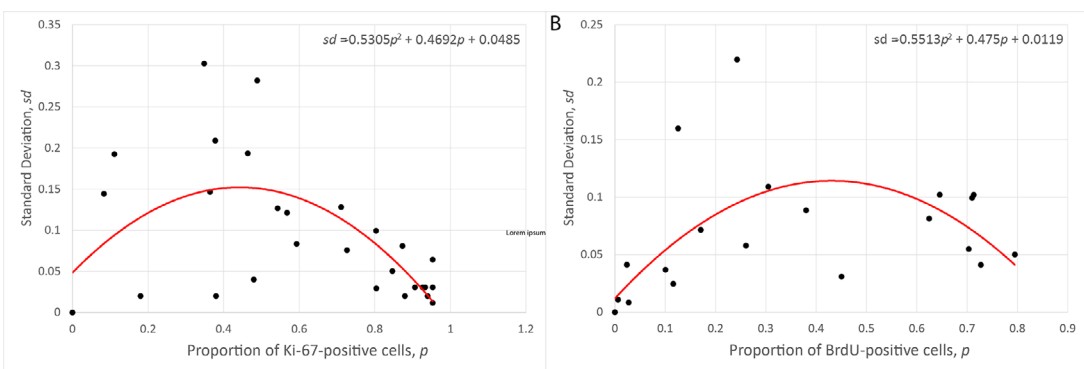

**Appendix 1—figure 3.** Relationship between the mean and the standard deviation of the proportion of Ki-67- (**A**) and BrdU-positive (**B**) cells observed at several crypt positions in three replicated control experiments.

## What-if analysis

We investigated the effect on the simulated crypt of increasing and decreasing the strength of the main signalling pathways, Wnt, BMP, and ZNRF3/RNF43 signalling, and modifying the Notch thresholds. For each alternative parameterization, except when decreasing ZNRF3/RNF43 signalling, the simulation was run for 30 d to ensure stability was reached with the new parameter set and the final 10 d were included in the analysis. When decreasing ZNRF3/RNF43 signalling, we simulated 60 d to demonstrate the expansion of the niche and analysed the final 10 d. The reference parameter set used as baseline was the ileal mouse crypt parameter set reported in *Appendix 1—table 1*. In all cases, we only consider modifications of one signalling mechanism at a time.

To study alternative Wnt signalling scenarios, we used the *WntRange* parameter (*Appendix 1—table 1*) to double and halve the spreading area of Wnt signals emitted by Paneth cells while we maintained the original *WntRange* value for Wnt-emitting mesenchymal cells at the bottom of the crypt (Section 1.7.1; *Appendix 1—figure 4A–F*). When *WntRange* was doubled, we observed increased number of stem and Paneth cells in a noticeably enlarged niche (*Appendix 1—figure 4C and D*), with cells choosing the stem cell fate instead of differentiating into absorptive progenitors. On the other hand, decreasing Wnt signalling, by halving *WntRange* in Paneth cells but maintaining its homeostatic value in mesenchymal cells, resulted in no apparent changes in niche cell composition (*Appendix 1—figure 4E and F*), which resembled published experimental results of persisting functional stem cells after Paneth cell ablation (*Durand et al., 2012*).

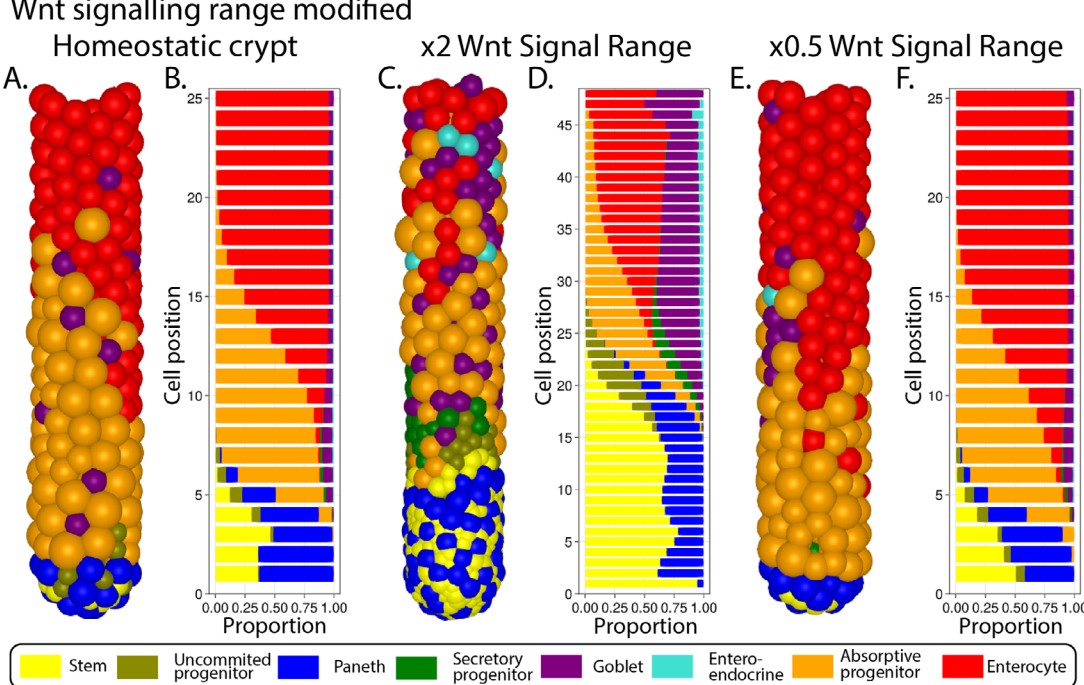

Wnt signalling range modified

**Appendix 1—figure 4.** Simulated ileal mouse crypt with modified Wnt signalling. Three-dimensional image (**A**) and cell composition (**B**) by position in homeostasis and likewise after doubling (**C, D**) and halving (**E, F**) Paneth cell generated Wnt signals, maintaining homeostatic levels of Wnt of mesenchymal sources.

The ZNRF3/RNF43-mediated negative feedback mechanism regulates the size of the niche by modulating Wnt signalling. We simulated increasing and decreasing the strength of the ZNRF3/RNF43 by doubling and halving, respectively, the parameter Z described in Section 1.7.2; *Appendix 1— figure 5A–F*. Following the increase in the intensity of ZNRF3/RNF43 signalling, we observed a decrease in the number of stem and Paneth cells, together with relatively minor changes in the transit-amplifying region (*Appendix 1—figure 5C and D*). On the other hand, when decreasing ZNRF3/RNF43 signalling levels, the niche expanded, resulting in a crypt dominated by Paneth and stem cells (*Appendix 1—figure 5E and F*) which replicates reported experimental phenotypes (*Koo et al., 2012*).

**Appendix 1—figure 5.** Simulated ileal mouse crypt with modified ZNRF3/RNF43 signalling. Three-dimensional image (**A**) and cell composition (**B**) by position in homeostasis and likewise after doubling (**C, D**) and halving (**E, F**) ZNRF3/RNF43 signalling strength.

To modify Notch signalling, we increased and decreased by 1 A.U. the Notch threshold required for lateral inhibition (*Appendix 1—figure 6A–F*). This Notch signalling threshold determines the number of contacting Notch-secreting cells (secretory lineage) required to inhibit the differentiation of stem cells into the secretory lineage. Thus, increasing this Notch threshold enhances the production of secretory cells leading to the increase in Paneth, goblet, and enteroendocrine cells (*Appendix 1—figure 6C and D*). Alternatively, decreasing the Notch threshold enhances differentiation into the absorptive lineage, reducing the number of Paneth and secretory cells (*Appendix 1—figure 6E and F*).

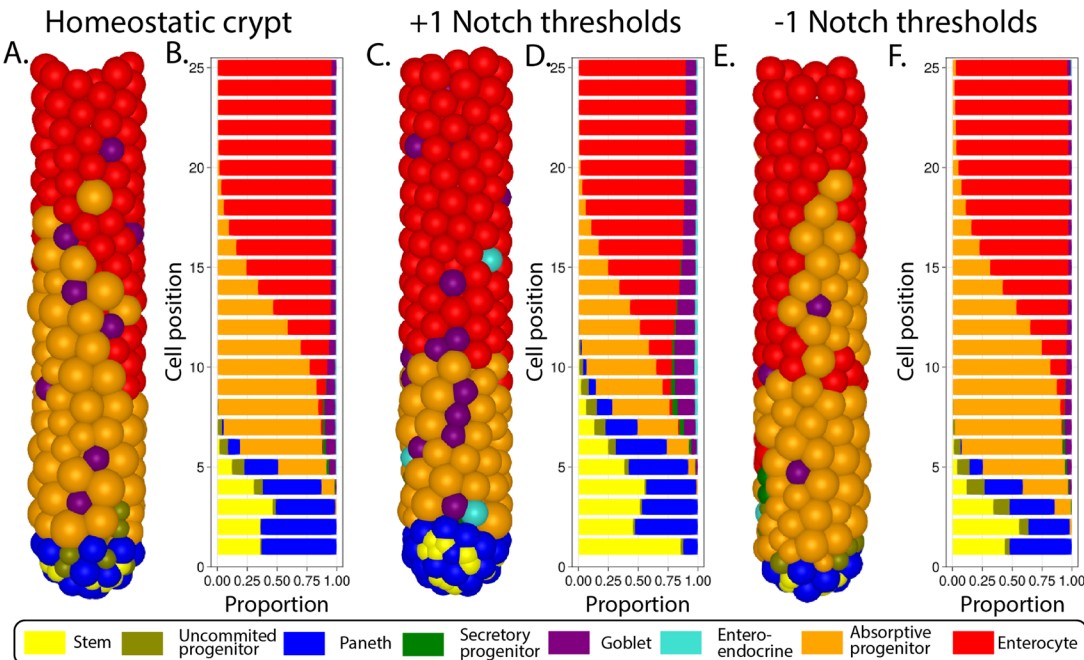

**Appendix 1—figure 6.** Simulated ileal mouse crypt with modified Notch signalling. Three-dimensional image (**A**) and cell composition (**B**) by position in homeostasis and likewise after increasing (**C, D**) and decreasing (**E, F**) the Notch threshold by 1 A.U.

We modified the range of diffusion of BMP signals by doubling and halving the parameter $A$ (*Appendix 1—figure 7A–F*) which denotes the amount of diffusing BMP signals, and hence affects the diffusion range, towards the base of the crypt (Section 1.7.4). When we increased the BMP signalling range, enterocytes differentiated at lower crypt positions, effectively reducing the transit-amplifying zone (*Appendix 1—figure 7A and B*). Decreasing BMP signalling strength by halving $A$ resulted in the increase in proliferative absorptive progenitors, which reach higher positions in the crypt (*Appendix 1—figure 7C and D*). The niche was largely unaffected in both cases (*Appendix 1—figure 7E and F*).

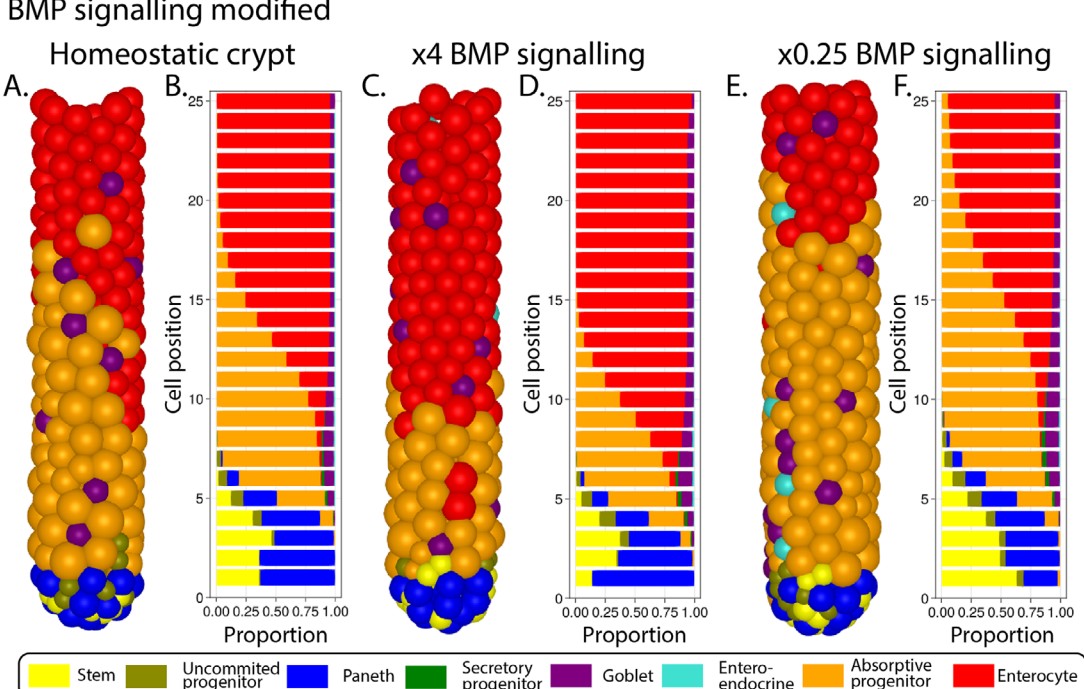

**Appendix 1—figure 7.** Simulated ileal mouse crypt with modified BMP signalling. Three-dimensional image (**A**) and cell composition (**B**) by position in homeostasis and likewise after a fourfold increase (**C, D**) and a fourfold decrease (**E, F**, respectively) of BMP signalling strength.

## Model implementation and parameterization

The model is implemented using the Julia programming language. The mechanical forces, cellular motion, and biochemical signalling are simulated with a fixed timestep of $dt = 0.0001$ d, while the proteins of the cell cycle model are simulated with a timestep of $0.00001$ d. Parameter values and means used for their identification are detailed in *Appendix 1—table 1*.

