## [Editor Report]

The proposed model makes an important contribution to the field, allowing a better understanding of the formation and response dynamics of the intestinal crypt through the effective evaluation of healthy, disease and treatment conditions. The authors provided convincing evidence of the validity of their model and their conclusions.

---

## [Decision Letter]

**Decision letter after peer review:**

Thank you for submitting your article "Homeostasis, injury and recovery dynamics at multiple scales in a self-organizing intestinal crypt" for consideration by *eLife*. Your article has been reviewed by 3 peer reviewers, including Mariana Gómez-Schiavon as Reviewing Editor and Reviewer #1, and the evaluation has been overseen by Aleksandra Walczak as the Senior Editor.

*Recommendations for the authors:*

– We recommend revising the writing in the introduction and the abstract to clarify the background, motivation, impact, and context of the research. We suggest that the authors explicitly state why studying the intestinal crypt is broadly important to the scientific community and why a mathematical model of it is necessary for progress in the development of oncotherapies. Further, we recommend highlighting the impact of their model of the intestinal crypt on the development of oncotherapies. Most of this information is included in the Discussion section, but not in the introduction or the abstract.

– ABM must be defined in the main text, not just the abstract.

– The authors claimed to demonstrate the model's application to epithelial research and drug discovery. They evaluate some known drugs, but we don't believe the term "discovery" is fully justified in the manuscript.

– We suggest that the authors specify how the model they developed compares to others available in the literature (references (15-18)) and its specific contribution. The authors mentioned that several agent-based models have been proposed to address the question of interest, but they failed to state what is the gap that their model is filling and in general how it differentiates from previous work. Given the comment on line 74, we can assume the novelty of the developed model lies in not including any "externally imposed behaviors", but this needs to be specified in the introduction.

– The importance of negative feedback loops in the signaling pathways is stated by the authors. Nevertheless, a clear diagram of the responsible regulatory networks is lacking. Such a diagram would allow the reader to clearly judge the presence and details of this or any other feedback in the networks in the context of the proposed conclusions. Additionally, this diagram would allow a clearer interpretation of how the multiple signaling pathways connect in the proposed model.

– As a suggestion for improving the readability of the manuscript, the third paragraph of the introduction (lines 46-55) seems disconnected from the rest of the text. Some rephrasing could help with the flux of the introduction.

– In lines 98 and 99, the authors claim "Specifically, in our model, high Wnt, and Notch signalling environments are required to maintain stemness." Is this statement supported experimentally?

– Often, it is not clear which statements are derived from experimental evidence versus model results. For example, in the sentence on lines 138-140, "a decreased number of enterocytes results in reduced production of BMP, which enables progenitor cells to divide and migrate further up the crypt before meeting BMP levels higher than the differentiation threshold".

– Regarding Figure 1B: The mathematical equations are not necessary. They're not in the main text of the paper. What do the green checkmark and red X mean?

– Regarding Figure 1C: Improperly labeled axes. What are numbers 0, 2, 3, 4, and 64 on the axes? What is p = 0.1 and p = 0.9? Some of these numbers are explained in the supplementary, but not in the main text of the paper.

– Regarding Figure 1D: What types of cells are in the light blue and dark green colors? Where did the authors obtain the data to generate this panel?

– Regarding Figure 2A-D: We find Figure 2A-D not particularly informative. We suggest the authors reevaluate its purpose and redesign it accordingly. For instance, if the purpose is to compare the cell cycle behavior between long and short cycles, panels B and D must be truly comparable, and the x-axis for both figures should be in equal proportions respective to the cell cycle lengths. If the purpose is to show the dependency of p27 over the cell cycle length, the values of p27 need to be shown as well, ideally including a diagram highlighting the interaction between p27 and the rest of the system. Additionally, each panel should have its own caption to increase clarity. Alternatively, the authors could plot the comparison between stem cells and transit-amplifying cells in the same panel and with fewer biochemical species.

– Regarding Figure 2E,F: The simulation and the data don't match. The conclusions drawn from these panels are questionable.

– We suggest rearranging the paragraph on lines 175-186: the purpose (why) of BrdU tracking and Ki-67 staining needs to be specified before explaining how it was done. In its current form, the paragraph can be very confusing for readers not familiar with the experimental technique.

– Some visualization aid (potentially in Figure 1A) would be helpful to understand retrograde movement defined as "when its velocity is negative in the z direction".

– Regarding Figure 3,4: It would be helpful to see side-by-side figures of healthy/homeostatic crypt organization vs the results of the ablation of stem cells or of CDK1 inhibition. Consider including adjacent panels or plotting on top of each other.

– Regarding Figure 3B: The red line for enterocytes appears in the top right corner.

– Regarding Figure 3D: Please use different colors to show directionality. Additionally, red and blue are used to indicate types of cells up to this point. And in general, it is really hard to interpret Figure 3D. Separating the bars (homeostasis vs cell ablation) might help.

– Figure 3E is never referenced in the main text, which is in general bad practice. As its goal is to be a comparison point with Figure 3F, we suggest putting both plots in one panel, each with a clear title.

– To be able to appreciate Figure 4A, we need a reference point, i.e. the case without CDK1 inhibition. The authors show this in a video, but we suggest including it as a figure here too.

– Regarding Figure 4C: Labels describing the cell cycle progress (as in Figure 2B,D) would be helpful.

– Regarding Figure 5: Each panel should have its own caption to increase clarity.

– Regarding Figure 5C: Recovery is after 144 hrs (yellow). That is more than two days (main text). Also, data vs experiments in the purple lines looks qualitatively different.

– Figure 5D: After 6, 24, 96, and 144 hrs, the proportion of proliferative cells is roughly the same. There is no recovery. Please explain why.

– Figure 5E-H: Not referenced in the main text of the paper. Probably not necessary to include.

– With exception of Figure 1, the quality of the figures is poor and can be easily improved. For instance, colors for plots Figure 2A-D can be better chosen to facilitate distinction between the lines (ideally considering colorblindness). We appreciate the attempt to keep the color associated with each cell type consistent through the plots, nevertheless in this case that just complicates the readability of the figures: there are too many cell types to keep track of, and some colors are too similar to each other, some colors are hard to see, there are too many lines in individual panels, and some of the designated colors are used in some other contexts. We suggest simplifying Figure 2A-C and similar plots to only show relevant lines, each with its own legend, and always starting with the y-axis in zero (note: the y-axis range in Figure 2A can be misleading, as it doesn't start in zero, but it isn't obvious at a first sight). Colors can then be chosen to help visualization in Figures 2E-F. (If some lines are excluded for readability in the panels, they can also be shown as figure supplements.)

– Clarify: Line 133 – How do you know what an optimal crypt cell composition is?

– Clarify: Line 149 – Why do you use the Csikasz-Nagy model? Clarify the main text and give some background. Additionally, what is the advantage of this model compared to others models that were described before? Could a similar model be used or was another model tested?

– Clarify: Line 176 – BrdU is field-specific knowledge. Please explain what it does to a more general audience.

– Clarify: Line 284 – What is partially published data?

– The full model could not be evaluated, as it is not available yet in BioModels (even if it appears as submitted). The authors should provide access to the reviewers for a full assessment.

– We suggest slowing down the cell movements/dynamics in all the videos. It is easy to follow the general dynamic of the crypt but a bit difficult to follow how single cells behave.

– A reminder in the video of which color represents each cell type in the crypt would help a lot to follow it.

– Clarify: Line 115 in the Supplementary – Why do the normal distributions have those means and variances? They're very specific.

– Clarify: Please clarify the model organism throughout the paper. Both mouse and human are referenced.

– Discussion: The potential applications of the model should be expanded in the discussion and/or response to reviewers. How exactly can this model be used in different intestinal diseases such as chronic inflammation?

– Discussion: In the last years 3D organoid technology has been a major advance in studying intestinal biology. Would it be possible to apply the model to study organoids' responses in vitro? And study the single-cell response using organoids?

– Discussion: It is well known that small intestinal crypts have a different spatial organization than a colonic crypt, and furthermore, in the small intestinal tissue there are differences among duodenum, ileum, and jejunum. Can this be predicted with the model?

– Discussion: Can the authors expand on how different drugs' actions can be predicted using the same model (e.g. 5 Fu compared to oxaliplatin)?

– Discussion: How close is the model to predicting how many cycles of chemotherapy should be given to a patient to eliminate cancer cells and not cause severe toxicity?

– Discussion: Could a knock-out mouse model of a specific pathway be used to validate the model?

---

## [Author Response]

Recommendations for the authors:– We recommend revising the writing in the introduction and the abstract to clarify the background, motivation, impact, and context of the research. We suggest that the authors explicitly state why studying the intestinal crypt is broadly important to the scientific community and why a mathematical model of it is necessary for progress in the development of oncotherapies. Further, we recommend highlighting the impact of their model of the intestinal crypt on the development of oncotherapies. Most of this information is included in the Discussion section, but not in the introduction or the abstract.

We have revised the abstract which now contains the following paragraph:

“The maintenance of the functional integrity of the intestinal epithelium requires a tight coordination between cell production, migration and shedding along the crypt-villus axis. Dysregulation of these processes may result in loss of the intestinal barrier and disease. With the aim of generating a more complete and integrated understanding of how the epithelium maintains homeostasis and recovers after injury, we have built a multi-scale agent-based model (ABM) of the mouse intestinal epithelium. We demonstrate that a stable self-organizing behaviour in the crypt emerges from the dynamic interaction of multiple signalling pathways, such as Wnt, Notch, BMP, RNF43/ZNRF3 and YAP-Hippo pathways, which regulate proliferation and differentiation, respond to environmental mechanical cues, form feedback mechanisms and modulate the dynamics of the cell cycle protein network.

The model recapitulates the crypt phenotype reported after persistent stem cell ablation and after the inhibition of the CDK1 cycle protein. Moreover, we simulated 5-fluorouracil (5-FU)-induced toxicity at multiple scales starting from DNA and RNA damage, which disrupts the cell cycle, cell signalling, proliferation, differentiation and migration and leads to loss of barrier integrity. During recovery, our in-silico crypt regenerates its structure in a self-organizing, dynamic fashion driven by dedifferentiation and enhanced by negative feedback loops. Thus, the model enables the simulation of xenobiotic-, in particular chemotherapy-, induced mechanisms of intestinal toxicity and epithelial recovery.

Overall, we present a systems model able to simulate the disruption of molecular events and its impact across multiple levels of epithelial organization and demonstrate its application to epithelial research and drug development.”

And we have rewritten the introduction which now includes the following paragraph

“The imbalance of this tightly orchestrated system contributes to pathological conditions, including microbial infections, intestinal inflammatory disorders, extra-intestinal autoimmune diseases, and metabolic disorders (Chelakkot et al, 2018). In addition, critically ill patients and patients receiving chemotherapy/radiotherapy often show severely compromised intestinal barrier integrity (Chelakkot et al., 2018). For instance, oncotherapeutics-induced gastrointestinal toxicity is frequently a life-threatening condition that leads to dose reduction, delay and cessation of treatment and presents a constant challenge for the development of efficient and tolerable cancer treatments (McQuade et al, 2016; Saltz et al, 2000; Saltz et al, 2001; Stein et al, 2010). This intestinal toxicity often results from the interaction of the drug with its intended molecular target such as cell cycle proteins (Zhang et al, 2021) or the disruption of the cycle through DNA-damage (Helleday et al, 2008). Multiscale models integrating our knowledge on how the epithelium maintains homeostasis and responds to injury can contribute to understand epithelial biology and to quantify the risk of intestinal toxicity during drug development.”

– ABM must be defined in the main text, not just the abstract.

We have amended the manuscript as suggested.

– The authors claimed to demonstrate the model's application to epithelial research and drug discovery. They evaluate some known drugs, but we don't believe the term "discovery" is fully justified in the manuscript.

All references to ‘drug discovery’ have been changed to ‘drug development’.

– We suggest that the authors specify how the model they developed compares to others available in the literature (references (15-18)) and its specific contribution. The authors mentioned that several agent-based models have been proposed to address the question of interest, but they failed to state what is the gap that their model is filling and in general how it differentiates from previous work. Given the comment on line 74, we can assume the novelty of the developed model lies in not including any "externally imposed behaviors", but this needs to be specified in the introduction.

A paragraph describing previous models has been added to the introduction and it is attached below. The main novel features of our model are described in detail in the discussion:

“Several agent-based models (ABMs) have been proposed to describe the complexity and dynamic nature of the intestinal crypt. Early models were used as in silico platforms to study the dynamics and cellular organisation of the crypt. For instance, one of the pioneering ABMs was used to study the distribution and organisation of labelling and mitotic indices (Meineke et al, 2001). This model comprises a fixed ring of Paneth cells beneath a row of stem cells, which divide asymmetrically to produce a stem cell and a transit-amplifying cell that terminally differentiates after a fixed number of divisions. Some subsequent models are lattice-free, recapitulate neutral drift of equipotent stem cells and describe proliferation and cell fate regulated by a fixed Wnt signalling spatial gradient, which is defined by the distance from the crypt base, with proliferating cells progressing through discrete phases of the cell cycle and showing variable duration of the G1 phase (Pitt-Francis et al, 2009). Further model refinements can be seen in the model of Buske et al. (2011), with stochastic cell growth and division time (Buske et al, 2011), Wnt levels defined by the fixed local curvature of the crypt and lateral inhibition driven by Notch signalling. Here, we present a lattice-free agent-based model that describes the spatiotemporal dynamics of single cells in the small intestinal crypt driven by the interaction of surface tethered Wnt signals, cell-cell Notch signalling, BMP diffusive signals, RNF43/ZNRF3-mediated feedback mechanisms and the cycle protein network responding to the crypt mechanical environment. We show that our computational model enables the simulation of the ablation and recovery of the stem cell niche as well as of how drug-induced molecular perturbations trigger a cascade of disruptive events spanning from the cell cycle to single cell arrest and/or apoptosis, altered cell migration and turnover and ultimately loss of epithelial integrity.”

Also we have rephrased the sentence “not including externally imposed behaviours”, which now reads as:

“with behaviours (proliferation, differentiation, fate decision, migration, etc.) determined largely by endogenous intracellular and intercellular interactions.”

– The importance of negative feedback loops in the signaling pathways is stated by the authors. Nevertheless, a clear diagram of the responsible regulatory networks is lacking. Such a diagram would allow the reader to clearly judge the presence and details of this or any other feedback in the networks in the context of the proposed conclusions. Additionally, this diagram would allow a clearer interpretation of how the multiple signaling pathways connect in the proposed model.

We have integrated a diagram showing the feedback regulatory loops into Figure 1A.

– As a suggestion for improving the readability of the manuscript, the third paragraph of the introduction (lines 46-55) seems disconnected from the rest of the text. Some rephrasing could help with the flux of the introduction.

We have re-worded this paragraph for a more logical flow, primarily by putting gut-relevant information first to establish why we focus on oncotherapeutics. The paragraph now reads:

“The imbalance of this tightly orchestrated system contributes to pathological conditions, including microbial infections, intestinal inflammatory disorders, extra-intestinal autoimmune diseases, and metabolic disorders (Chelakkot et al., 2018). In addition, critically ill patients and patients receiving chemotherapy/radiotherapy often show severely compromised intestinal barrier integrity (Chelakkot et al., 2018). For instance, oncotherapeutics-induced gastrointestinal toxicity is frequently a life-threatening condition that leads to dose reduction, delay and cessation of treatment and presents a constant challenge for the development of efficient and tolerable cancer treatments (McQuade et al., 2016; Saltz et al., 2000; Saltz et al., 2001; Stein et al., 2010). This intestinal toxicity often results from the interaction of the drug with its intended molecular target such as cell cycle proteins (Zhang et al., 2021) or the disruption of the cycle through DNA-damage (Helleday et al., 2008). Multiscale models integrating our knowledge on how the epithelium maintains homeostasis and responds to injury can contribute to understand epithelial biology and to quantify the risk of intestinal toxicity during drug development.”

– In lines 98 and 99, the authors claim "Specifically, in our model, high Wnt, and Notch signalling environments are required to maintain stemness." Is this statement supported experimentally?

A supporting reference with experimental evidence has been added to this sentence.

Tian et al. Nature 2011 Vol. 478 Issue 7368 Pages 255-9*.*

– Often, it is not clear which statements are derived from experimental evidence versus model results. For example, in the sentence on lines 138-140, "a decreased number of enterocytes results in reduced production of BMP, which enables progenitor cells to divide and migrate further up the crypt before meeting BMP levels higher than the differentiation threshold".

We have added “*in our model*” here and throughout the manuscript to clarify modelling assumptions/hypotheses vs experimental data when needed.

– Regarding Figure 1B: The mathematical equations are not necessary. They're not in the main text of the paper. What do the green checkmark and red X mean?

We have revised Figure 1B and equations have been removed. The ‘Green checkmark and red X’ marks are omitted from the new diagram explaining Notch inhibition. Also new diagrams explain the Wnt, BMP and ZNRF3 feedback loops.

– Regarding Figure 1C: Improperly labeled axes. What are numbers 0, 2, 3, 4, and 64 on the axes? What is p = 0.1 and p = 0.9? Some of these numbers are explained in the supplementary, but not in the main text of the paper.

We have removed all numbers from the Notch and Wnt signalling axes and indicated the direction of signalling increase with “Low” and “High”. All parameter values can be found in the Appendix and Table 1. This includes references to probability of fate differentiation, *p*, and threshold values for Wnt, BMP and Notch axis.

Wnt/BMP axis: replaced with description of values: 64 is replaced with ‘*Fate Determinant Wnt Level*’ and 4 with ‘*BMP and Wnt level for Differentiation’*.

Notch axis: removed numerical values, replaced with ‘Low’ and ‘High’ Notch.

The Figure caption has been amended as:

“C) Cell fate determination. High Wnt signalling and activation of Notch are required to maintain stemness. Low Notch signalling determines differentiation into secretory fates, including Paneth cells in high Wnt signalling regions, or goblet/enteroendocrine progenitors in low Wnt regions. Absorptive progenitors develop from stem cells in low Wnt conditions and divide 3-5 times, before becoming terminally differentiated when Wnt signal levels are decreased and cells find sufficient BMP signals;”

– Regarding Figure 1D: What types of cells are in the light blue and dark green colors? Where did the authors obtain the data to generate this panel?

Colours (of enteroendocrine = light blue, secretory progenitor = dark green) in 1A are now amended to be identical to 1D. The same colour code is now used for all plots.

Panel 1D shows the average cell composition, per cell position, of a simulated crypt in homeostasis. The data is generated by the model and shows a good agreement with reported experimental data in (Buske et al., 2011).

We have clarified the Figure caption that now reads:

D) Average composition of a simulated healthy/homeostatic crypt (over 100 simulated days), showing the relative proportion of cells at each position.

and the manuscript text now reads:

“… our model describes single cells that generate and respond to signals and mechanical pressures in the crypt-villus geometry to give rise to a self-organizing crypt which has stable spatial cell composition over time (Figure 1D) and reproduces reported experimental data (Buske et al., 2011).”

– Regarding Figure 2A-D: We find Figure 2A-D not particularly informative. We suggest the authors reevaluate its purpose and redesign it accordingly. For instance, if the purpose is to compare the cell cycle behavior between long and short cycles, panels B and D must be truly comparable, and the x-axis for both figures should be in equal proportions respective to the cell cycle lengths. If the purpose is to show the dependency of p27 over the cell cycle length, the values of p27 need to be shown as well, ideally including a diagram highlighting the interaction between p27 and the rest of the system. Additionally, each panel should have its own caption to increase clarity. Alternatively, the authors could plot the comparison between stem cells and transit-amplifying cells in the same panel and with fewer biochemical species.

We reduced 4 plots, A-D, to 2 plots (A and B). The new plots (A and B) show the dynamics of the main cycle proteins, including p27, in cells with long (figure A) and short (figure B) cycle in a comparable manner, with equal x-axis. We highlighted in the figure caption that a diagram of the protein interaction network, including the interaction of p27 with the rest of the proteins, can be found in the original paper of Csikasz-Nagy, where this model was first described. We did not bring here the diagram due to its size and complexity.

– Regarding Figure 2E,F: The simulation and the data don't match. The conclusions drawn from these panels are questionable.

Figure 2E-F are now Figures 2C-D. As per response to one of the main comments, we have performed several adjustments that include, refinement of the counting algorithm and of the Ki67 and BrdU staining simulations and addition of an estimation of the experimental error to the simulated responses. A description of these changes is described in a new section in the appendix called “ABM simulation of Ki-67 and BrdU Staining”

With these changes we think we have achieved a more satisfactory agreement between observed and predicted results and updated all figures with Ki67 and BrdU staining simulated results.

However, despite our efforts, we did not reach a perfect agreement between BrdU staining simulated and observed results. The simulation behind our predictions is a complex process describing multiple dynamic processes interacting at different scales. Some of our model assumptions, i.e duration of S-phase, dynamics of BrdU integration into the DNA…, may be too simplistic for these datasets and/or our model maybe be missing some important piece of information on BrdU staining process which is not currently understood. This is difficult to identify at the moment, but we are committed to keep refining our models as new knowledge emerges to enhance their performance and application. On the other hand, BrdU staining involves a considerably large experimental error inherent to both large biological and technical variability (sampling region and sampling time, counting, reagents…). Other authors have also encountered difficulties to obtain a good match between simulated and observed BrdU staining results (see Buske et al, 2011).

– We suggest rearranging the paragraph on lines 175-186: the purpose (why) of BrdU tracking and Ki-67 staining needs to be specified before explaining how it was done. In its current form, the paragraph can be very confusing for readers not familiar with the experimental technique.

We have added a new section in the appendix called “ABM simulation of Ki-67 and BrdU Staining” with further details of these simulations and extended the description of these experiments in the following paragraph of the main manuscript.

“To demonstrate the performance of the model to reproduce the spatiotemporal cell dynamics and composition of a homeostatic crypt, we simulated previous published mouse experiments (Parker et al, 2017; Parker et al, 2019) comprising 5-bromo-29-deoxyuridine (BrdU) tracking (Figure 2E) and Ki-67 staining (Figure 2F). BrdU, is a thymidine analogue often used to track proliferative cells and their descendants along the crypt–villus axis (Gratzner, 1982; Nowakowski et al, 1989). BrdU is incorporated into newly synthesized DNA of dividing cells during the S-phase and transmitted to daughter cells, regardless of whether they proliferate. If the exogenous administration of this molecule is discontinued, the cell label content is diluted by each cell division and is no longer detected after 4–5 generations (Wilson et al, 2008). To simulate the BrdU chase experiment after a single BrdU pulse, we assumed that any cell in S-phase incorporated BrdU permanently into its DNA for the first 120 minutes after injection of BrdU and BrdU cell content was diluted upon cell division such that after five cell divisions, BrdU was not detectable. See Appendix for a complete description. The BrdU chase simulation showed that the observed initial distribution of cells in S-phase as well as division, differentiation and migration of BrdU-positive cells over time were replicated by our model (Figure 2E).

Ki-67 is produced by actively proliferating cells during the S-, G2- and M-phases of the division cycle (Sobecki et al, 2017). Ki-67 is detected after exiting the cycle (Miller et al, 2018) and during G1 in continuously cycling cells (Sobecki et al., 2017), which is related to the time required for the protein to be catabolized (Miller et al., 2018). Our simulations assumed that Ki-67 is detected in continuously cycling cells, cells re-entering the cycle after arrest except during G1, as well as in differentiated cells that were cycling within the past 6 hours and recently drug-arrested cells. See Appendix for a complete description. Similarly, we observed that the ABM-simulated spatial distribution along the crypt of Ki-67 positive cells recapitulated observations in mouse ileum (Figure 2F).”

– Some visualization aid (potentially in Figure 1A) would be helpful to understand retrograde movement defined as "when its velocity is negative in the z direction".

We have added arrows showing the direction of regular cellular migration and retrograde motion to Figure 1A. We have attached a low resolution version of Figure 1 here with this change highlighted in a red box.

– Regarding Figure 3,4: It would be helpful to see side-by-side figures of healthy/homeostatic crypt organization vs the results of the ablation of stem cells or of CDK1 inhibition. Consider including adjacent panels or plotting on top of each other.

We have included an image of a simulated 3D crypt in homeostasis in Figure 4 (CDK1 inhibition plot), because that figure is less crowded. In addition, in this figure, we have extended the simulation duration to achieve full recovery of the crypt after ablation and highlighted in the caption that homeostatic cell counts for comparison can be seen after recovery.

– Regarding Figure 3B: The red line for enterocytes appears in the top right corner.

We have divided the plots to show with clarity all cell lineages in the crypt and villus during stem cell ablation and recovery.

– Regarding Figure 3D: Please use different colors to show directionality. Additionally, red and blue are used to indicate types of cells up to this point. And in general, it is really hard to interpret Figure 3D. Separating the bars (homeostasis vs cell ablation) might help.

We have amended this figure as suggested. The bars corresponding to homeostasis and ablation scenarios are now plotted side-by-side (separate) and represented by different colours. Dark/light colour shows upward and downward motion. We have also clarified the legend to improve understandability.

– Figure 3E is never referenced in the main text, which is in general bad practice. As its goal is to be a comparison point with Figure 3F, we suggest putting both plots in one panel, each with a clear title.

We have corrected this typo, the original sentence referred to Figure 3D instead of 3E, which is now referenced in the main text. As suggested these plots are now combined into one panel and explanatory text has been added into the plots.

– To be able to appreciate Figure 4A, we need a reference point, i.e. the case without CDK1 inhibition. The authors show this in a video, but we suggest including it as a figure here too.

As suggested, we have included an image of a simulated 3D crypt in homeostasis side-by-side with the injured crypt. We have also extended the duration of the simulation to achieve homeostasis and highlighted in the caption that homeostatic cell counts and cell cycle protein dynamics can be seen after recovery for comparison.

– Regarding Figure 4C: Labels describing the cell cycle progress (as in Figure 2B,D) would be helpful.

The labels are now added and colours have been modified to match those in other cell cycle figures.

– Regarding Figure 5: Each panel should have its own caption to increase clarity.

Captions have been addressed as suggested.

– Regarding Figure 5C: Recovery is after 144 hrs (yellow). That is more than two days (main text). Also, data vs experiments in the purple lines looks qualitatively different.

The total number of crypt cells is recovered by day 2 after 5-FU high dose treatment is interrupted, however the crypt cell composition is still affected with increased proliferative cells resulting from the cell proliferation enhancing feedback mechanisms responding to injury. We have clarified this in the following section of the manuscript:

“Figure 5C shows that predicted and observed Ki-67 positive cells declined gradually over time at all positions in the crypt during the 5-FU high dose treatment but the numbers recovered, reaching values above baseline, two days after the interruption of 5-FU administration. The increased rebound of the proliferative crypt compartment after the treatment was captured in our ABM by the implemented BMP-mediated feedback mechanism from mature enterocytes to proliferative cells (see BMP signalling section in the Appendix).”

We made several adjustments to improve model predictions: we refined the counting algorithm that determines cell position and improved the Ki67 and BrdU staining simulations by altering the simulated staining criteria and adding of an estimation of the experimental error to the simulated responses. A description of these changes is described in a new section in the appendix called “*ABM simulation of Ki-67 and BrdU Staining*”. With these changes we think we have achieved a more satisfactory agreement between observed and predicted results and updated all figures with Ki67 and BrdU staining simulated results.

– Figure 5D: After 6, 24, 96, and 144 hrs, the proportion of proliferative cells is roughly the same. There is no recovery. Please explain why.

Figure 5D describes the proportion of BrdU^+^ cells during 5-FU low dose treatment. This treatment had a a minimal impact on crypt proliferation and on the number of cells on the epithelium. The plots have been moved to the appendix.

– Figure 5E-H: Not referenced in the main text of the paper. Probably not necessary to include.

These plots including the response to 5-FU low dose have now been moved to the Appendix while new plots describing the response of the cell cycle protein network to the drug have been added to Figure 5 (now Figures 5A and B).

– With exception of Figure 1, the quality of the figures is poor and can be easily improved. For instance, colors for plots Figure 2A-D can be better chosen to facilitate distinction between the lines (ideally considering colorblindness). We appreciate the attempt to keep the color associated with each cell type consistent through the plots, nevertheless in this case that just complicates the readability of the figures: there are too many cell types to keep track of, and some colors are too similar to each other, some colors are hard to see, there are too many lines in individual panels, and some of the designated colors are used in some other contexts. We suggest simplifying Figure 2A-C and similar plots to only show relevant lines, each with its own legend, and always starting with the y-axis in zero (note: the y-axis range in Figure 2A can be misleading, as it doesn't start in zero, but it isn't obvious at a first sight). Colors can then be chosen to help visualization in Figures 2E-F. (If some lines are excluded for readability in the panels, they can also be shown as figure supplements.)

We have simplified the plots in the manuscript main figures to increase clarity and improve line visualization. We have also added new Supplementary Figures S1-S2 with all cell lineages in logarithmic scale. Line colours are changed according to the Seaborn’s colourblind palette.

– Clarify: Line 133 – How do you know what an optimal crypt cell composition is?

All references to optimal cell composition have been changes to ‘homeostatic cell composition’.

– Clarify: Line 149 – Why do you use the Csikasz-Nagy model? Clarify the main text and give some background. Additionally, what is the advantage of this model compared to others models that were described before? Could a similar model be used or was another model tested?

To build the cell cycle in our ABM an exhaustive search of cell cycle models was performed with the following criteria: 1) include the core cell cycle proteins (cyclins, CDKs, Wee1, p21/p27), 2) comprise a variable, such as the mass variable of the Csikasz-Nagy model, that we could link to cell features in our ABM, such as the cell size and 3) include sufficient mechanistic detail to allow the description of drug-cell cycle interactions. The Csikasz-Nagy model fulfils all these criteria and was available in the BioModels repository, with accessible, correct code. It is also a well-established model that builds upon the seminal work of Novak and Tyson on the dynamics of cell cycle protein network. In addition, through experimentation, we found this model particularly amenable and robust to constant, dynamic alterations of the parameters required to dynamically vary the length of the cell cycle according to contact inhibition in our ABM.

Other versions of this model could also be integrated in our ABM and a more sophisticated version could be implemented in the future if required to simulate other mechanisms of cycle protein network disruption. Other models with other hypotheses governing protein dynamics remain to be proven.

We have clarified this in the following paragraph in the appendix:

“We have used the model of Csikasz-Nagy et al. (Csikasz-Nagy et al, 2006), that recreates the mammalian cell cycle and is available in Biomodels (Le Novère & Csikasz-Nagy, 2006). This model is an extension of the seminal work of Novak and Tyson that help reveal the complex nonlinear dynamics of the cell cycle proteins (Novak et al, 2001; Novak & Tyson, 1993, 2004). The Csikasz-Nagy model provides multiple necessary features such as core cell cycle proteins, a mass variable that can be coupled to the volume of the single cells in our ABM and sufficient mechanistic detail to enable a detailed description of drug-cycle interactions.”

– Clarify: Line 176 – BrdU is field-specific knowledge. Please explain what it does to a more general audience.

The following paragraph detailing the use of BrdU has been extended in the main text.

“BrdU, is a thymidine analogue often used to track proliferative cells and their descendants along the crypt–villus axis (Gratzner, 1982; Nowakowski et al., 1989). BrdU is incorporated into newly synthesized DNA of dividing cells during the S-phase and transmitted to daughter cells, regardless of whether they proliferate. If the exogenous administration of this molecule is discontinued, the cell label content is diluted by each cell division and is no longer detected after 4–5 generations (Wilson et al., 2008).”

– Clarify: Line 284 – What is partially published data?

That term has been removed form the text.’ Further clarification has been added in material and methods, which now reads as follows:

“We used BrdU tracking and Ki-67 immunostaining data from previously published experiments in healthy mice (Parker et al., 2017; Parker et al., 2019) and following 5-FU treatment (Jardi et al, 2022). The samples from this later study (Jardi et al., 2022) were analysed again to count Ki-67 positive cells at each position along the longitudinal crypt axis, for 30-50 individual hemi crypt units per tissue section per mouse as previously described (Williams et al, 2016).”

– The full model could not be evaluated, as it is not available yet in BioModels (even if it appears as submitted). The authors should provide access to the reviewers for a full assessment.

Model files are provided with this reply. These are ‘.jl’ files for use with Julia. The model (the files provided with this reply) will be freely publicly available through BioModels upon acceptance of this manuscript for publication.

– We suggest slowing down the cell movements/dynamics in all the videos. It is easy to follow the general dynamic of the crypt but a bit difficult to follow how single cells behave.

All videos have been recreated with slower cell movement and a higher frame rate to allow easier tracking of cell behaviour.

– A reminder in the video of which color represents each cell type in the crypt would help a lot to follow it.

In the new version of the videos we have added a fixed legend.

– Clarify: Line 115 in the Supplementary – Why do the normal distributions have those means and variances? They're very specific.

We have expanded that section to justify the choice of parameters for the normal distributions as follows:

“When a proliferative cell is created, it is assigned a desired final size, rfinal=2 3r∗, where r∗∼N(0.35,0.00875) for stem cells and r∗∼N(0.5,0.0125) for all other cells. The mean values, 0.5 and 0.35, of the radius of progenitor and stem cells, respectively, were determined for an average, non-proliferative or proliferative progenitor cell to have, without loss of generality, a diameter of 1 while the diameter of an average stem cell is slightly smaller, 0.7. In this way, the model captures the smaller size described for columnar LGR5+ stem cells (Barker *et al*, 2008), which additionally helps recapitulate the mechanics and cell composition of the niche. The variance of the radius was determined by our implementation of the cell cycle model in the ABM. In our model, the volume of the cell is equated to the cell’s mass parameter of the Csikasz-Nagy model and, hence, the cell final radius determines the duration of the cell cycle as described above. By simulating the cell cycle model, we observed that large values of the standard deviation resulted in some cells progressing through the cycle too quickly and, therefore, failing to complete the cell cycle correctly. This analysis provided an upper limit to the coefficient of variation (CV) = 0.025 to ensure all cells progress regularly through the cycle during homeostasis. This results in values of the standard deviation of the radius of 0.0125 and 0.00875 for progenitor cells and stem cells, respectively. Of note, a cell radius CV of 0.025 corresponds to a cell volume CV of about 0.075 which is not far from the reported experimental CV for cell volume, about 0.11 (Bell & Anderson, 1967).”

– Clarify: Please clarify the model organism throughout the paper. Both mouse and human are referenced.

We have clarified the model organism throughout the text.

– Discussion: The potential applications of the model should be expanded in the discussion and/or response to reviewers. How exactly can this model be used in different intestinal diseases such as chronic inflammation?

In the text, we discussed three applications of our model for the study of epithelial biology by simulating stem cell ablation and to assess epithelial injury associated with two common mechanisms of action of oncotherapeutics, DNA damage and cell cycle protein network disruption. In principle, the implemented inter- and intra-cellular signalling mechanisms can be modified to simulate the effects of different drugs and diseases. Moreover, the model can be developed further to include additional molecular targets by adding new pathways or expanding molecular networks. In the discussion, we have modified the following section to emphasise this, that reads:

“One of the important applications of our modelling approach lies in the development of safer oncotherapeutics. The model enables the prediction of intestinal injury associated with efficacious dosing schedules in order to minimize toxicity while maintaining efficacy of investigational drugs. We demonstrated the application of our model to predict potential intestinal toxicity phenotypes induced by CDK1 inhibition as well as to describe the disruption of the epithelium at multiple scales triggered by RNA and DNA damage leading to the loss of integrity of the intestinal barrier and diarrhea following 5-FU treatment. The drug-induced perturbation of other cell cycle proteins or signalling pathways, already integrated into the model, is straightforward to simulate with the current version of the model while the resolution of molecular networks can be increased, or new pathways incorporated into the ABM, to describe additional drug mechanisms of action.”

For diseases that involve exclusively the disruption of the epithelium (burns, physical erosion…) our model alone may be sufficient if assuming no extra medical complications. However, for modelling inflammatory diseases, such as IBD, or infections, the role of the mucosal immune system, microbiome and other elements such as the mesenchymal and enteric nervous system needs to be carefully assessed and included in the model when relevant.

– Discussion: In the last years 3D organoid technology has been a major advance in studying intestinal biology. Would it be possible to apply the model to study organoids' responses in vitro? And study the single-cell response using organoids?

Yes organoids and ABMs are mutually supportive of each other. ABMs can be used to test hypotheses behind organoid responses and organoids can provide single cell data suitable to develop ABMs with precision. Moreover, we are actively researching on the application of organoids combined with modelling to predict clinical GI toxicity in patients. If successful, this work will form the basis of a full manuscript of its own.

To capture this point in the manuscript we have added the following section:

“While most of the crypt biology understanding integrated in our model derives from mouse epithelial studies, human-derived intestinal organoids and microphysiological systems, now routinely used in research, can provide highly precise information at the single cell level to inform ABM development. In return, ABMs can help test hypotheses behind organoid responses in health and disease conditions.”

– Discussion: It is well known that small intestinal crypts have a different spatial organization than a colonic crypt, and furthermore, in the small intestinal tissue there are differences among duodenum, ileum, and jejunum. Can this be predicted with the model?

We focussed on the small intestine because of its higher proliferative activity and hence relevance to study oncotherapeutics-induced toxicity but also because of the richness and abundance of datasets.

In the current the model, we assumed that differences between jejunum and ileum of the small intestine, can be implemented using the physical parameters of the crypt (height, circumference) and the length-scale of BMP signals to regulate cell spatial organization. The different parametrization is included in the Parameter Table 1 and was based on data obtained in the different tissues. Likewise, we are actively working on modelling the colonic crypt by modifying the parametrization accordingly to the current understanding of the colonic crypt biology, e.g. Paneth cells replaced by deep crypt secretory cells, larger physical dimensions, slower proliferation and distinctive cell spatial organization resulting from different parametrization of Wnt, BMP and RNF43/ZNRF3 signalling pathways.

– Discussion: Can the authors expand on how different drugs' actions can be predicted using the same model (e.g. 5 Fu compared to oxaliplatin)?

The model is set up to describe perturbation of DNA synthesis and various cell cycle proteins as well as the main signalling pathways. Other molecular targets need to be implemented in the model, (but the model was designed to be modular and easily modifiable).

For 5-FU we have modelled FUTP-induced DNA and RNA damage in proliferative cells leading to cell apoptosis and cycle arrest. In the case of oxaliplatin, the kinetics of platinum-DNA adducts in each proliferative cell leading to cell apoptosis will likely need to be integrated into the model.

We added a section to the manuscript (to address an earlier question) that explains further how to use our model to simulate other drug MoA and we bring here:

“Drug-induced perturbation of other cell cycle proteins or signalling pathways, already integrated into the model, are straightforward to simulate with the current version of the model while the resolution of pathways can be increased, or new molecular networks incorporated into the ABM to describe additional drug mechanisms of action.”

– Discussion: How close is the model to predicting how many cycles of chemotherapy should be given to a patient to eliminate cancer cells and not cause severe toxicity?

Mitigating GI toxicity is one of the main applications to drug development of our model. We are already using this model to predict intestinal injury associated with efficacious dosing schedules in order to minimize toxicity while maintaining efficacy of investigational drugs. Mechanistic models are best placed to generate predictions prior to clinical trials and, hence, maximize patients’ safety.

We have added the following section in the manuscript:

“One of the important applications of our modelling approach lies in the development of safer oncotherapeutics. The model enables the prediction of intestinal injury associated with efficacious dosing schedules in order to minimize toxicity while maintaining efficacy of investigational drugs.”

– Discussion: Could a knock-out mouse model of a specific pathway be used to validate the model?

Yes, in fact knowledge on epithelial biology, such as cell fate and signalling regulation, gathered using knock-out mouse experiments has already been instrumental to develop and validate our model. For instance, the new section on sensitivity analysis in the supplementary material shows the reaction of the model to changes in signalling mechanisms and recreates the response of knockout mouse models of the chosen pathways. Another example is the phenotype associated to ZNRF3 knockout which was revealed in knock out experiment and it is now in the text:

“These premises imply that Paneth cells enhance their own production by generating Wnt signals and inducing prolonged division times, which increases stem and Paneth cell production and could lead to unlimited expansion of the niche, recapitulating the phenotype seen in RNF43/ZNRF3 knockout mice (Koo et al, 2012).”

Knock out mouse models are invaluable tools to study epithelial biology and acquire the deep understanding required to develop ABM models. We will continue to refine our model by adding newly discovered behaviours of the epithelium with this or other research tools.

References

Barker N, van de Wetering M, Clevers H (2008) The intestinal stem cell. *Genes Dev* 22: 1856-1864

Bell GI, Anderson EC (1967) Cell growth and division. I. A mathematical model with applications to cell volume distributions in mammalian suspension cultures. *Biophys J* 7: 329-351

Buske P, Galle J, Barker N, Aust G, Clevers H, Loeffler M (2011) A comprehensive model of the spatio-temporal stem cell and tissue organisation in the intestinal crypt. *PLoS Comput Biol* 7: e1001045

Chelakkot C, Ghim J, Ryu SH (2018) Mechanisms regulating intestinal barrier integrity and its pathological implications. *Exp Mol Med* 50: 1-9

Csikasz-Nagy A, Battogtokh D, Chen KC, Novak B, Tyson JJ (2006) Analysis of a generic model of eukaryotic cell-cycle regulation. *Biophys J* 90: 4361-4379

Gratzner HG (1982) Monoclonal antibody to 5-bromo- and 5-iododeoxyuridine: A new reagent for detection of DNA replication. *Science* 218: 474-475

Helleday T, Petermann E, Lundin C, Hodgson B, Sharma RA (2008) DNA repair pathways as targets for cancer therapy. *Nat Rev Cancer* 8: 193-204

Jardi F, Kelly C, Teague C, Fowler-Williams H, Rodrigues D, Jo H, ., Ferreria S, Herpers B, Van Heerden M, de Kok TM *et al* (2022) Mouse organoids as an in vitro tool to study the in vivo intestinal response to cytotoxicants. *Archives of Toxicology (Accepted)*

Koo B-K, Spit M, Jordens I, Low TY, Stange DE, van de Wetering M, van Es JH, Mohammed S, Heck AJR, Maurice MM *et al* (2012) Tumour suppressor RNF43 is a stem-cell E3 ligase that induces endocytosis of Wnt receptors. *Nature* 488: 665-669

Le Novère N, Csikasz-Nagy A, 2006. Cell Cycle Model BioModels.

McQuade RM, Stojanovska V, Donald E, Abalo R, Bornstein JC, Nurgali K (2016) Gastrointestinal dysfunction and enteric neurotoxicity following treatment with anticancer chemotherapeutic agent 5-fluorouracil. *Neurogastroenterol Motil* 28: 1861-1875

Meineke FA, Potten CS, Loeffler M (2001) Cell migration and organization in the intestinal crypt using a lattice-free model. *Cell Prolif* 34: 253-266

Miller I, Min M, Yang C, Tian C, Gookin S, Carter D, Spencer SL (2018) Ki67 is a Graded Rather than a Binary Marker of Proliferation versus Quiescence. *Cell reports* 24: 1105-1112.e1105

Novak B, Pataki Z, Ciliberto A, Tyson JJ (2001) Mathematical model of the cell division cycle of fission yeast. *Chaos* 11: 277-286

Novak B, Tyson JJ (1993) Numerical analysis of a comprehensive model of M-phase control in Xenopus oocyte extracts and intact embryos. *J Cell Sci* 106 ( Pt 4): 1153-1168

Novak B, Tyson JJ (2004) A model for restriction point control of the mammalian cell cycle. *J Theor Biol* 230: 563-579

Nowakowski RS, Lewin SB, Miller MW (1989) Bromodeoxyuridine immunohistochemical determination of the lengths of the cell cycle and the DNA-synthetic phase for an anatomically defined population. *J Neurocytol* 18: 311-318

Parker A, Maclaren OJ, Fletcher AG, Muraro D, Kreuzaler PA, Byrne HM, Maini PK, Watson AJ, Pin C (2017) Cell proliferation within small intestinal crypts is the principal driving force for cell migration on villi. *FASEB J* 31: 636-649

Parker A, Vaux L, Patterson AM, Modasia A, Muraro D, Fletcher AG, Byrne HM, Maini PK, Watson AJM, Pin C (2019) Elevated apoptosis impairs epithelial cell turnover and shortens villi in TNF-driven intestinal inflammation. *Cell Death Dis* 10: 108

Pitt-Francis J, Pathmanathan P, Bernabeu MO, Bordas R, Cooper J, Fletcher AG, Mirams GR, Murray P, Osborne JM, Walter A *et al* (2009) Chaste: A test-driven approach to software development for biological modelling. *Comput Phys Commun* 180: 2452-2471

Saltz LB, Cox JV, Blanke C, Rosen LS, Fehrenbacher L, Moore MJ, Maroun JA, Ackland SP, Locker PK, Pirotta N *et al* (2000) Irinotecan plus fluorouracil and leucovorin for metastatic colorectal cancer. Irinotecan Study Group. *N Engl J Med* 343: 905-914

Saltz LB, Douillard JY, Pirotta N, Alakl M, Gruia G, Awad L, Elfring GL, Locker PK, Miller LL (2001) Irinotecan plus fluorouracil/leucovorin for metastatic colorectal cancer: a new survival standard. *Oncologist* 6: 81-91

Sobecki M, Mrouj K, Colinge J, Gerbe F, Jay P, Krasinska L, Dulic V, Fisher D (2017) Cell-Cycle Regulation Accounts for Variability in Ki-67 Expression Levels. *Cancer Res* 77: 2722-2734

Stein A, Voigt W, Jordan K (2010) Chemotherapy-induced diarrhea: pathophysiology, frequency and guideline-based management. *Ther Adv Med Oncol* 2: 51-63

Williams JM, Duckworth CA, Vowell K, Burkitt MD, Pritchard DM (2016) Intestinal Preparation Techniques for Histological Analysis in the Mouse. *Curr Protoc Mouse Biol* 6: 148-168

Wilson A, Laurenti E, Oser G, van der Wath RC, Blanco-Bose W, Jaworski M, Offner S, Dunant CF, Eshkind L, Bockamp E *et al* (2008) Hematopoietic stem cells reversibly switch from dormancy to self-renewal during homeostasis and repair. *Cell* 135: 1118-1129

Zhang M, Zhang L, Hei R, Li X, Cai H, Wu X, Zheng Q, Cai C (2021) CDK inhibitors in cancer therapy, an overview of recent development. *Am J Cancer Res* 11: 1913-1935